# FGCaMP7, an Improved Version of Fungi-Based Ratiometric Calcium Indicator for In Vivo Visualization of Neuronal Activity

**DOI:** 10.3390/ijms21083012

**Published:** 2020-04-24

**Authors:** Natalia V. Barykina, Vladimir P. Sotskov, Anna M. Gruzdeva, You Kure Wu, Ruben Portugues, Oksana M. Subach, Elizaveta S. Chefanova, Viktor V. Plusnin, Olga I. Ivashkina, Konstantin V. Anokhin, Anna V. Vlaskina, Dmitry A. Korzhenevskiy, Alena Y. Nikolaeva, Konstantin M. Boyko, Tatiana V. Rakitina, Anna M. Varizhuk, Galina E. Pozmogova, Fedor V. Subach

**Affiliations:** 1Laboratory for Neurobiology of Memory, P.K. Anokhin Research Institute of Normal Physiology, 125315 Moscow, Russia; n.barykina@nphys.ru (N.V.B.); ivashkina_oi@nrcki.ru (O.I.I.); k.anokhin@gmail.com (K.V.A.); 2Institute for Advanced Brain Studies, M.V. Lomonosov Moscow State University, 119991 Moscow, Russia; vsotskov@list.ru (V.P.S.); gruzdeva@neuro.mpg.de (A.M.G.); 3Complex of NBICS Technologies, National Research Center “Kurchatov Institute”, 123182 Moscow, Russia; subach_om@nrcki.ru (O.M.S.); witkax@mail.ru (V.V.P.); annavlaskina@yandex.ru (A.V.V.); igra-voina@yandex.ru (D.A.K.); nikolaeva_ay@nrcki.ru (A.Y.N.); taniarakitina@yahoo.com (T.V.R.); 4Sensorimotor Control Research Group, Max Planck Institute of Neurobiology, 82152 Martinsried, Germany; youkwu@neuro.mpg.de (Y.K.W.); rportugues@neuro.mpg.de (R.P.); 5Institute of Neuroscience, Technical University of Munich, 80802 Munich, Germany; 6Munich Cluster for Systems Neurology (SyNergy), Munich, Germany; 7Department of NBIC-technologies, Moscow Institute of Physics and Technology, 123182 Moscow, Russia; yelizaveta.chefanova@phystech.edu; 8Bach Institute of Biochemistry, Research Center of Biotechnology of the Russian Academy of Sciences, 119071 Moscow, Russia; kmb@inbi.ras.ru; 9Laboratory of Hormonal Regulation Proteins, M.M. Shemyakin and Yu.A. Ovchinnikov Institute of Bioorganic Chemistry of the Russian Academy of Sciences, 117997 Moscow, Russia; 10Department of Biophysics, Research and Clinical Center of Physical-Chemical Medicine of Federal Medical Biological Agency, 119435 Moscow, Russia; annavarizhuk@gmail.com (A.M.V.); pozmge@gmail.com (G.E.P.); 11Department of Biophysics, Center for Precision Genome Editing and Genetic Technologies for Biomedicine, 119435 Moscow, Russia

**Keywords:** calcium imaging, genetically encoded calcium indicator, protein engineering, crystal structure, FGCaMP7, FGCaMP

## Abstract

Genetically encoded calcium indicators (GECIs) have become a widespread tool for the visualization of neuronal activity. As compared to popular GCaMP GECIs, the FGCaMP indicator benefits from calmodulin and M13-peptide from the fungi *Aspergillus niger* and *Aspergillus fumigatus*, which prevent its interaction with the intracellular environment. However, FGCaMP exhibits a two-phase fluorescence behavior with the variation of calcium ion concentration, has moderate sensitivity in neurons (as compared to the GCaMP6s indicator), and has not been fully characterized in vitro and in vivo. To address these limitations, we developed an enhanced version of FGCaMP, called FGCaMP7. FGCaMP7 preserves the ratiometric phenotype of FGCaMP, with a 3.1-fold larger ratiometric dynamic range in vitro. FGCaMP7 demonstrates 2.7- and 8.7-fold greater photostability compared to mEGFP and mTagBFP2 fluorescent proteins in vitro, respectively. The ratiometric response of FGCaMP7 is 1.6- and 1.4-fold higher, compared to the intensiometric response of GCaMP6s, in non-stimulated and stimulated neuronal cultures, respectively. We reveal the inertness of FGCaMP7 to the intracellular environment of HeLa cells using its truncated version with a deleted M13-like peptide; in contrast to the similarly truncated variant of GCaMP6s. We characterize the crystal structure of the parental FGCaMP indicator. Finally, we test the in vivo performance of FGCaMP7 in mouse brain using a two-photon microscope and an NVista miniscope; and in zebrafish using two-color ratiometric confocal imaging.

## 1. Introduction

The development of genetically encoded calcium indicators (GECIs) is a rapidly progressing area, providing tools that enable the monitoring of a wide range of processes in live cells and animals, including neuronal brain activity. To date, several types of GECIs have been developed. The first type consists of two GFP-like fluorescent proteins (FPs) fused by a Ca^2+^-binding protein, typically calmodulin (CaM) or troponin C (TnC) [1,2]. This sensor type relies on the Förster resonance energy transfer (FRET) mechanism, which is a result of the overlapping between emission and excitation spectra of two fluorophores. Upon calcium ion binding, the FRET signal of this type of sensor increases. Another type of sensor consists of a single circularly permutated FP sandwiched between a CaM/M13 pair [3,4,5]. The intensity of these intensiometric indicators varies with changes in the chromophore environment upon calcium binding-induced complex formation between CaM and M13-like peptide. We recently made great efforts in the development of another type of GECI, by insertion of a calcium-binding domain. This type of GECI includes the NTnC-like family, which contains truncated TnC from *Opsanus tau* inserted into mNeonGreen or mEYFP FPs [6,7,8]. The NTnC-like family preserves the mechanism of fluorescence change common to single FP-based sensors but also possesses advantages such as a reduced number of Ca^2+^-binding sites and small molecular size of the indicator. The ncpGCaMP6s and NCaMP7 indicators contain an insertion of the CaM/M13-peptide pair in EGFP and mNeonGreen proteins, respectively [9,10].

Among GECIs, the intensiometric single-FP based GCaMP family is the most popular, which has been used in experiments in vivo with updated variants from GCaMP2 to jGCaMP7 [4,5,11,12]. However, extensive evidence has been accumulated, showing that the chronic expression of GCaMPs leads to such unexpected side-effects as cytotoxicity, cell death, and abnormal neuronal excitability. Thus, the unregulated expression of GCaMP2 in mouse heart cells resulted in significant cardiomegaly in adult transgenic mice, similar to CaM overexpression [13]. Chronic expression of the GCaMP3 calcium indicator caused cytotoxicity and death of neurons in mice [12,14]. These undesired side-effects are probably connected with abnormal GCaMPs accumulation in the nuclei of neurons, which has also been observed for GCaMP6s, which is commonly used in vivo [4,15]. The spontaneous fluorescence transients of such nuclear-filled neurons had long decay times and the neurons had reduced calcium changes [12]. Moreover, there exist functional data concerning aberrant epileptiform activity in some genotypes of transgenic mice expressing GCaMP6 [16]. The length and amplitude of spikes in this aberrant activity is substantially higher than the activity observed in unaffected mice and, thus, can complicate the interpretation of studies obtained in such mice. Finally, it has been shown that GCaMP might perturb the gating of the Ca_v_1/CaM complex in neurons, which is crucial for excitation–transcription coupling and, therefore, may affect downstream Ca_v_1-signaling pathways [15].

The data collected to date argues for a strong need to search for ways to overcome the side-effects caused by the overexpression of the GECIs in neurons. One of the most promising ways to address the problem of intracellular environment perturbation by calmodulin in GECIs is the application of troponin C (from muscle) as a Ca^2+^-binding motif for the development of GECIs. This approach has been used for the development of double- and single-FP-based GECIs [2,6,7,8,17]. Another approach is the introduction of additional “protective” motif from the IQ domain of neuromodulin fused to CaM in its apo indicator state. Several variants of indicators based on previously developed GCaMP3, GCaMP5G, and GCaMP6 have been implemented with “protective” motifs and named GCaMPs-X [15]. Such apoCaM binding motifs in GCaMPs-X efficiently protect apoCaM from interfering with Ca_v_1 channels, such that neurons expressing these indicators exhibit normal morphology. However, this strategy increases the molecular size of the indicator and introduces an additional domain, which may perturb the intracellular environment. Moreover, it does not address the problem of intracellular environment perturbation by the Ca^2+^-bound indicator. Changing the topology of the indicator can also reduce the interaction of the GECI with the intracellular environment. FRAP experiments for the NCaMP7 indicator (based on the insertion of the CaM/M13-peptide into mNeonGreen) demonstrated that, when the topology is changed in this way, calcium indicator interactions with the intracellular environment may, to some extent, be prevented [9]. 

Several ratiometric single-FP based calcium indicators have been developed. Ratiometric-pericam was created based on circularly-permuted EYFP fused to an M13-peptide derived from the CaM-binding region of the skeletal muscle myosin light-chain kinase and mammalian CaM [18]. It had a bimodal excitation spectrum peaking at 415 and 494 nm, and the relative intensity of green fluorescence (at 511 and 517 nm) at 494 and 415 nm excitation was changed by about 10-fold between Ca^2+^-saturated and Ca^2+^-free forms. The GEX-GECO1 ratiometric green GECI based on GFP fused to mammalian M13-peptide and CaM had a fluorescence ratio change of 18–26-fold [3,19]. However, both forms of GEX-GECO1 have substantially lower brightness than EGFP. Two series of Y-GECO1 and Y-GECO2 ratiometric indicators have been developed using mPapaya (a monomeric variant of the *Zoanthus species* yellow FP), mammalian M13-peptide, and CaM [20,21]. However, these indicators had an inverse response for anionic form and dim large Stokes shift fluorescence (about 3–12-fold dimmer than EGFP fluorescence). 

Most of the developed calcium indicators employ CaM or troponin C from metazoa (theleosts, birds, and mammals) as Ca^2+^-binding proteins [2,11,17]. To increase the diversity of Ca^2+^-binding motifs applied in GECIs and to minimize the side-effects of GECIs overexpression in cells, we recently developed a ratiometric FGCaMP indicator and its FGCaMP2–FGCaMP4 mutants based on EGFP FP and CaM/M13 Ca^2+^-binding domains from *Aspergillus* fungi with amino acid sequences different from homologous sequences of mouse CaM and M13-peptide (85% and 40% amino acid identity for CaM and M13-peptide, respectively, between *Aspergillus niger/Aspergillus fumigatus* and *Mus musculus*) [19]. FGCaMP has high mobility in mammalian cells at low calcium concentrations, as compared to GCaMP6s and G-GECO1 indicators [19], which is beneficial in terms of its lack of interaction with intracellular proteins. However, FGCaMP has a bi-phasic shape of dependence of fluorescence on calcium ion concentration, has less ΔF/F response during neuronal activity (as compared to GCaMP6s), the basis of its sensitivity to calcium ions has not been studied, and it has not been tested for applicability in two- and one-photon calcium imaging in mouse brains.

To address these limitations, we obtained the crystal structure of the FGCaMP indicator, to better understand the basis underlying its functioning. Then, we developed a novel indicator FGCaMP7 which possesses the ratiometric phenotype of its progenitor, but with substantially improved response and calcium affinity in vitro. We characterized spectral, biochemical, and kinetic properties of FGCaMP7 in detail, as well as characterizing the response of FGCaMP7 to changes in calcium concentrations in cultured mammalian cells and rodent neuronal cultures during spontaneous or induced calcium activity. FGCaMP7 demonstrated increased ΔR/R_0_ response in both non-stimulated and stimulated neuronal cultures, as compared to its FGCaMP progenitor. We revealed the inertness of the FGCaMP7 indicator variant with deleted M13-peptide when expressed in mammalian cells, in contrast to a similarly truncated version of GCaMP6s. We expressed FGCaMP7 in the mouse brain and characterized its response in vivo using two-photon microscopy and a head-mounted NVista miniature microscope. Finally, FGCaMP7 was successfully applied for two-color ratiometric calcium imaging in zebrafish in vivo. 

## 2. Results and Discussion

### 2.1. Structural Characterization of FGCaMP Calcium Indicator

To understand the structural basis underlying FGCaMP functioning and to search for potential mutation hotspots, we determined the crystal structure of the original FGCaMP calcium indicator [19] in its Ca^2+^-bound state at 3.2 Å resolution (Appendix A). Like all GCaMP-like calcium indicators, according to its crystal structure, FGCaMP represents two domains: a cpEGFP moiety and a CaM/M13-peptide complex with four bound calcium ions (Figure 1a, Appendix A). The cpEGFP moiety contains 11 β-strands forming a β-barrel with chromophore formed by TYG^184-186^ buried inside the barrel. CaM includes 8 α-helixes organized into four calcium-binding motifs (EF1–EF4). Each motif co-ordinates a single calcium ion by six contacts, as typical for calcium-binding domains (Appendix A) [22]. The M13-like peptide is tightly bound to the complex CaM-4 Ca^2+^. The CaM/M13 pair is linked to the cpEGFP moiety by two linkers (with a length of 3 and 2 amino acids, respectively) with chromophore partially exposed from the cpEGFP barrel (see Figure 1a, Appendix A). 

Synthetic GFP chromophore in neutral water solution has an absorption maximum at 370 nm and an absorption maximum at 470–500 nm in GFP-like fluorescent proteins has been observed due to the surrounding amino acid environment inside the protein [23,24]. We have previously shown that the ratiometric indicator FGCaMP has one main absorption peak at 493 nm in Ca^2+^-bound state and one absorption peak at 402 nm in Ca^2+^-free state (Table 1 and [19]). To understand which amino acid residues participate in the stabilization of the FGCaMP anionic form with an absorption maximum at 493 nm in Ca^2+^-bound state, we analyzed the amino acid environment of the FGCaMP chromophore using the obtained FGCaMP crystal structure and studied the impacts of the suggested point mutations on the FGCaMP absorption spectrum. According to the structural data analysis, we mutated the amino acid residues of FGCaMP located on the β-barrel of cpEGFP (N23, F39, I41, V77, S79, V180, T181, Q213, R215), on CaM (M267), and on linker 2 (L266), which were within 3.4–4.8 Å of the chromophore and which potentially affect the ratio of protonated and anionic forms of FGCaMP (Appendix A). Two additional positions of the linkers (L21 and T265) were also mutated but were located far from the chromophore surroundings. Substitutions to small, non-polar residues (F39G, I41G, V77G, S79A, V180G, T181G, Q213A, R215A, L266G, and M267G) or to structurally similar non-polar or polar residues (L21M, N23L, S79C, and T265S) were made. Mutants were expressed in *Escherichia coli* and proteins were extracted from bacteria and dialyzed against buffer containing 10 mM CaCl_2_. To determine which substitutions stabilized the 493 nm-absorbing form, the ratios of forms with absorption maxima at 402 and 493 nm (Abs^402^/Abs^493^) were estimated.

First, we estimated the impacts of the introduced mutations on spectral characteristics and the Abs^402^/Abs^493^ absorption ratio. The absorption peaks of protonated/anionic forms for FGCaMP/S79C, FGCaMP/L266G, and FGCaMP/R215A mutants in Ca^2+^-saturated state were observed at 408/498 nm, 398/500 nm, and 403/485 nm, respectively; while, for other mutants, absorbance peaks for both forms were around 405/490 nm, with a slight shift in the range of 3 nm (Appendix A). The Abs^402^/Abs^493^ absorption ratios were changed to the greatest extent for three FGCaMP mutants—FGCaMP/S79C, FGCaMP/L266G, and FGCaMP/R215A—and were 22-, 16-, and 12-fold higher, compared to the FGCaMP indicator, respectively (Appendix A). The Q213A, N23L, and S79A substitutions had Abs^402^/Abs^493^ ratios increased by 2.9-, 3.0-, and 1.9-fold, respectively, compared to the respective ratio for FGCaMP. In FGCaMP/F39G, FGCaMP/V77G, and FGCaMP/T181G mutants, the formation of the anionic 493 nm-absorbing form was blocked, making it impossible to evaluate the absorbance ratio (Appendix A). Five other substitutions (L21M, I41G, V180G, T265S, and M267G) did not change the absorbance ratio significantly, compared to the ratio for the control FGCaMP (Appendix A). Thus, according to the impacts of FGCaMP mutations on the Abs^402^/Abs^493^ ratios, they were divided into four groups: mutations significantly increasing the Abs^402^/Abs^493^ ratio (S79C, R215A, and L266G), mutations moderately increasing the Abs^402^/Abs^493^ ratio (N23L, S79A, and Q213A), mutations with absorption only at 402 nm (F39G, V77G, and T181G), and mutations with no effect on the Abs^402^/Abs^493^ ratio (L21M, I41G, V180G, T265S, and M267G).

Next, we attempted to explain the observed impacts of the mutations with a pronounced effect on the FGCaMP absorption. Among the eight amino acid residues which affected the Abs^402^/Abs^493^ ratio, S79 (S205 in GFP) was located at the largest distance from the chromophore (4.44 Å) (Appendix A) but had the most pronounced impact on the Abs^402^/Abs^493^ ratio (Appendix A). In the case of the wild-type GFP, S205 is known to participate in the hydrogen bonding network from E222 (E96 in FGCaMP, see Appendix A) through the water molecule to the phenolic hydroxyl group of the chromophore, thus maintaining the protonated form of GFP [25]. As the S–H group is more acidic than O–H [26], we suggest that cysteine in position 79 protonates to the hydroxyl of the FGCaMP chromophore and stabilizes the protonated state of the chromophore. S79A mutation also stabilized the protonated form of the chromophore, perhaps due to the small size of the alanine residue which increased the solvent accessibility of the chromophore, but with a more moderate effect on the Abs^402^/Abs^493^ ratio compared to the S79C mutation (Appendix A). 

The L266 residue, on the linker between cpEGFP and CaM, was at a distance of 3.70 Å from the phenolic hydroxyl of the chromophore (Appendix A). Its substitution to the small non-polar glycine changed the Abs^402^/Abs^493^ ratio 16-fold, probably as a consequence of the decreased hydrophobic core near the chromophore, which may have led to an increase in chromophore solution accessibility and, subsequently, to stabilization of protonated form of the FGCaMP/L266G mutant (Appendix A). 

Another mutation, R215A (R96 in GFP), also increased the Abs^402^/Abs^493^ ratio, indicating that this position was important for stabilization of the FGCaMP anionic form. R215 provided a positive charge at 2.77 Å distance from the carbonyl oxygen of imidazolinone (Appendix A), which may stabilize the anionic form of FGCaMP. This hypothesis is also supported by the data obtained previously for the GFP protein, showing that R96 stabilizes the anionic form of the mature chromophore [28]. It has been described earlier that the R96 position in GFP is important for chromophore cyclization and the red-shift in the excitation maximum of intact protein (compared to the denatured one) and the R96A mutation in GFP blue-shift excitation maximum of the protein from 489 to 468 nm [29]. Accordingly, in the case of FGCaMP, the R215A mutation also blue-shifted the excitation maximum of the anionic form from 493 to 485 nm (Appendix A).

The Q213A mutation (position Q94 in GFP) slightly increased the Abs^402^/Abs^493^ ratio, probably due to the removal of hydrogen bond (H-bond) donors/acceptors near the imidazolinone ring of the chromophore (Appendix A). Therefore, the H-bond donor/acceptors in position 213 and positive charge in position R215 are important for stabilization of the FGCaMP anionic form in Ca^2+^-saturated state, due to the support of the polar chromophore environment; similar to that in GFP [30]. T181 (T62 in GFP) and Q213 residues in FGCaMP create a polar environment (Appendix A), stabilizing the anionic 493 nm-absorbing form of FGCaMP, as has been shown earlier for GFP [30,31]. Probably for this reason, the T181G mutation lacked a 493 nm absorbing form and showed only a protonated 402 nm absorbing form (Appendix A). The R215A and T181G mutations also disturbed protein folding with efficient chromophore formation. The mutation of N23 to non-polar leucine, at a 3.90 Å distance from the chromophore (Appendix A), led to a moderate increase in the Abs^402^/Abs^493^ ratio; probably due to destabilization of the anionic form of the FGCaMP chromophore via deletion of the direct H-bond with the hydroxyl of the phenolic group of the chromophore (Appendix A).

The F39G mutation blocked formation of the anionic form of the FGCaMP indicator (Appendix A). F39 (F165 in GFP) seemed to create a non-polar environment around the chromophore and hold the polar core (composed of R215 and Q213 residues) around the chromophore (Appendix A), as has been earlier predicted for GFP [30]. Furthermore, the F39 residue may provide hydrophobicity and participate in stacking interactions with the chromophore, thus stabilizing the anionic form of FGCaMP. 

Threonine 203 in GFP forms an H-bond with the phenolic hydroxyl of the chromophore and, so, the T203I mutation led to the absence of the 475 nm absorbing form in the wild-type GFP [31]. The T203V substitution, which cannot form an H-bond with the phenolic hydroxyl of the chromophore, is also known to destabilize the anionic form of GFP [32]. However, even though FGCaMP has valine in the 77 position (analogous to the 203 position in GFP), the anionic 493 nm absorbing form in FGCaMP dominated over the protonated 402 nm absorbing one (Appendix A). We suggest that, in the case of FGCaMP, the V77 residue forms a hydrophobic core around the chromophore together with other non-polar amino acids; this core may be important for the stabilization of the FGCaMP anionic form. V77G mutation led to the disappearance of anionic 493 nm absorbing form of FGCaMP, probably due to a reduction of the size of hydrophobic core (Appendix A) and increasing solvent accessibility for the chromophore. Both F39G and V77G mutations impaired FGCaMP folding with efficient chromophore formation (Appendix A). 

Hence, of the 12 mutants of the FGCaMP indicator near the chromophore (i.e., within 3.4–4.8 Å) plus the two mutants with substitutions in linker positions 9–10 Å far from the chromophore, we found three mutations (S79C, R215A, and L266G) which substantially affected the absorption ratio for both protonated and anionic forms of the FGCaMP indicator in Ca^2+^-saturated state. Three other mutations (F39G, V77G, and T181G) blocked the formation of anionic form of FGCaMP. Hence, the residues in the positions 39, 77, 79, 181, 215, and 266 participate in the stabilization of the anionic form of the FGCaMP indicator. We suggest that the hydrophobic core around chromophore is important for the stabilization of anionic form. Namely, the mutations L266G, F39G, and V77G reduced (or fully eliminated) formation of anionic form, probably by decreasing the hydrophobic core around chromophore (Appendix A). The polar amino acids near the hydroxyl of a phenolic ring of the chromophore (S79 and N23) or close to the carbonyl group of imidazolinone of the chromophore (Q213A, R215A, and T181G) also substantially contributed to the stabilization of the anionic form of FGCaMP (Appendix A).

### 2.2. Development of Improved Version of GECI Based on Fungi Calcium-Binding Parts

Introducing the mutations described above within the surroundings of the FGCaMP chromophore, we tried to find variants with properties outperforming the initial FGCaMP indicator. To that end, we conducted a contrast analysis of purified mutants for both 402- and 493 nm- absorbing forms (Appendix A). In the case of 493 nm-absorbing forms, nine mutants had contrasts 1.4–11.3-fold smaller than the contrast of FGCaMP. FGCaMP/L21M, FGCaMP/N23L, FGCaMP/V180G, FGCaMP/Q213A, and FGCaMP/T265S mutants had similar contrasts and mutant FGCaMP/S79A had a contrast 1.8-fold higher than the contrast of FGCaMP, but the contrast of its 402-nm absorbing form was 23-fold smaller than the contrast of FGCaMP. The 402-nm absorbing forms of all other mutants had contrasts 1.8–9.9-fold smaller than the contrast of the respective form of FGCaMP (Appendix A). Hence, none of the structurally guided or rationally introduced mutations could enhance the properties of the FGCaMP indicator at no cost for the other characteristics.

As we were not able to improve the contrast of the FGCaMP indicator by introducing mutations in the surroundings near its chromophore, we decided to use the strategy of random mutagenesis and screening for further indicator evolution. The main drawback of FGCaMP is its bi-phasic Ca^2+^-binding curve with high- (K_d1_) and low-affinity (K_d2_) calcium components at 493 nm excitation, which ensures a non-linear response of the indicator with elevated of calcium ion concentration, potentially substantially reducing the fluorescence response of the indicator at low calcium ion concentrations in vivo. To further improve the properties of the FGCaMP indicator (Figure 1a, top), we chose mutants which we developed previously by directed mutagenesis of CaM/M13-peptide [19] with a mainly mono-phasic calcium-binding curve as a consequence of decreased contrast of the second (K_d2_) component at 493 nm excitation. These mutants had widely varying K_d_s values (FGCaMP2/T28D, K_d1_ 433 nM at 493 nm; FGCaMP3, K_d1_ 208 nM at 493 nm; FGCaMP4, K_d1_ 98 nM at 493 nm; Appendix A, Appendix A). Based on these mutants, we generated three random libraries. We analyzed mutants from these libraries and collected clones with the highest brightness and K_d1_ contrast at both 402 nm and 493 nm excitation and decreased K_d2_ contrast at 493 nm excitation, using two-step screening strategy in *E. coli* cells [7,8,19].

Clones selected after the screening of libraries based on FGCaMP2/T28D and FGCaMP3 had K_d1_ around 2.5 and 1.5 μM, respectively, which are not appropriate for the monitoring of calcium neuronal activity (K_d_ of 200 nM is considered optimal for neurons). The most promising variants were found in the library based on FGCaMP4 and, therefore, we focused on the directed evolution of this variant. The FGCaMP4 library was analyzed using a two-step screening strategy in each round of random mutagenesis. During the first step, we performed imaging of the indicator’s library targeted to the *E. coli* periplasm on Petri dishes and selected clones with the highest fluorescence ratio before and after treatment, using a buffer that contained ethylenediaminetetraacetic acid (EDTA) using 405/40BP and 480/40BP nm excitation filters and 510LP and 535/40BP nm emission filters, respectively. During the second step, the selected clones were analyzed in bacterial extracts in B-Per reagent. To eliminate the second component with low affinity to calcium ions, we selected proteins with minimal fluorescence ratio at 10 mM Ca-EGTA ([Ca^2+^ = 39 μM]) to 9 mM Ca-EGTA ([Ca^2+^ = 1.35 μM]) at 490 nm excitation. As a result, after three rounds of mutagenesis, we found an FGCaMP4.03 clone (Appendix A, Appendix A) with monophasic Ca^2+^-binding curve and 300 nM affinity to calcium ions in the presence of 1 mM MgCl_2_ and fluorescence contrast 1.6-fold higher than the contrast of original FGCaMP at 493 nm in vitro (23.3 ± 1.2 for FGCaMP4.03 vs. 14.7 ± 0.6 for FGCaMP). The selected mutant had approximately 2-fold lower brightness, compared to the original FGCaMP, and slow association/dissociation kinetics in spontaneously active cultured neurons. Hence, we decided to further improve the selected FGCaMP4 variant. 

After eight rounds of the random mutagenesis and screening described above, we found three promising variants—named FGCaMP5, FGCaMP6, and FGCaMP7—which preserved the ratiometric response of the parental FGCaMP to calcium ions and had in vitro fluorescence contrasts to 10 mM CaCl_2_ addition of 8.0-, 9.0-, and 10.1-fold at 400 nm excitation and 17.6-, 33.0-, and 32.7-fold at 498 nm excitation, respectively (Appendix A). 

FGCaMP5 and FGCaMP7 had 19 amino acid substitutions relative to the original FGCaMP, and FGCaMP6 had 20 amino acid substitutions (Appendix A). Among these mutations, 13 or 12 were located in the *A. niger/A. fumigatus*-derived M13-peptide/CaM of FGCaMP5 and FGCaMP6 or FGCaMP7, respectively; four or five mutations were located in the cpEGFP-based fluorescent domain of FGCaMP5 or FGCaMP6 and FGCaMP7, respectively; and two mutations were located in linkers between the Ca^2+^-binding and fluorescent domains (Appendix A). All mutations located in fluorescent domain were external to the β-barrel of GFP and were not likely to affect chromophore properties (Appendix A). All three chosen indicators had five mutations in the Ca^2+^-binding EF-hands: three mutations in EF2, and two in EF3. N327D, N364D, and S368D substitutions were previously introduced by site-directed mutagenesis; these mutations have been shown to decrease the K_d_ value of high-affinity components for the FGCaMP indicator [19]. FGCaMP5 had one unique N15I mutation in the M13-peptide, FGCaMP6 had two unique T162A and T272A substitutions, and FGCaMP7 had one unique K88Q mutation in the fluorescent domain (Appendix A). The location of the FGCaMP7 mutations along the gene of FGCaMP are presented schematically in Figure 1a (bottom).

### 2.3. In Vitro Properties of Improved Purified FGCaMP Variants

The key in vitro characteristics of the improved variants of the FGCaMP indicator were determined and are summarized in Table 1 and Appendix A. Spectroscopic data showed that FGCaMP5, FGCaMP6, and FGCaMP7 exhibited absorbance peaks at 400 nm in Ca^2+^-free state and 496–498 nm in Ca^2+^-saturated state (Appendix A and Figure 1b,c). When excited at 400 and 496–498 nm, the FGCaMP variants fluoresced at one emission peak at 516 nm (Appendix A and Figure 1b,c). All three FGCaMP variants had slightly (3–5 nm) smaller Stokes shift, compared to the original FGCaMP. The 400 and 496–498 nm absorption maxima can be attributed to the protonated and anionic forms of GFP chromophore, respectively, similar to the original FGCaMP indicator [19,23]. In Ca^2+^-saturated states, the brightness of FGCaMP5, FGCaMP6, and FGCaMP7 were 1.37-, 1.13-, and 1.16-fold higher than the brightness of the FGCaMP anionic form, respectively (Appendix A). In Ca^2+^-free states, the brightness of FGCaMP5 and FGCaMP7 were 1.66- and 1.23-fold higher than that of FGCaMP, respectively; while the 400-nm absorbing form of FGCaMP6 had only 65% brightness of the original FGCaMP indicator (Appendix A).

Upon calcium ion binding, the maximal fluorescence of FGCaMP7 and FGCaMP6 in the presence of 1 mM MgCl_2_ (equivalent to 0.58–1 mM free Mg^2+^ ion concentration, resembling that of 0.5–1.5 mM in the cytosol of mammalian cells [33,34]) showed 10.1- and 9.0-fold decreases and 32.7- and 33.0-fold increases at 400 and 496–498 nm excitation, respectively (Table 1 and Appendix A). These contrasts were 1.4- and 1.3-fold and 2.1- and 2.2-fold higher than the fluorescence contrasts of the respective forms of the original FGCaMP indicator. The fluorescence contrasts for the FGCaMP5 mutant (8.0 ± 0.4- and 17.6 ± 1.1-fold at 400 and 498 nm, respectively) were comparable to those of FGCaMP (7.1 ± 0.2- and 15.3 ± 0.5-fold at 402 and 493 nm, respectively). Thus, the selected FGCaMP variants are excitation-ratiometric calcium indicators with 140–330-fold maximal fluorescence contrast ratio change upon calcium ion binding. The maximal fluorescence contrast ratio for FGCaMP7 (330-fold) was 3.0-fold higher than that for FGCaMP (109-fold) in the presence of 1 mM MgCl_2_. Recently, another ratiometric calcium indicator series, Y-GECO2s, has been published with maximal contrasts ratio 1.1–2.3-fold higher than the contrast ratio for the FGCaMP7 indicator, which demonstrated the largest contrast among the studied FGCaMPs [21]. However, Y-GECO2s still suffer from low extinction coefficients and its molecular brightness at 412 and 522 nm excitation were 2.8–3.3-fold and 3.1–3.5-fold lower than the brightness of the FGCaMP indicators at 400 and 498 nm excitation, respectively [21]. Hence, enhanced FGCaMPs are the GECIs of choice among the currently available ratiometric indicators, in terms of high dynamic range and molecular brightness. 

We next assessed the affinity of FGCaMP variants to Ca^2+^ ions. According to equilibrium binding titration curves, FGCaMP demonstrates bi-phasic Ca^2+^ binding in the range of 0–39 μM free Ca^2+^ at 493 nm excitation. Such bi-phasic Ca^2+^ binding of FGCaMP is a limitation, as it decreases the sensitivity of the indicator to low calcium ion concentration transients and violates its linear response at high calcium ion concentrations when expressed in neurons. In the presence of 1 mM MgCl_2_, FGCaMP5, FGCaMP6, and FGCaMP7 demonstrated mono-phasic calcium titration curves with K_d_ values of 98 ± 3 nM, 350 ± 13 nM, and 240 ± 6 nM, respectively (Figure 1d and Appendix A; Table 1 and Appendix A). These values were 4.7-, 1.3-, and 1.9-fold lower than the K_d_ value for FGCaMP under the same conditions and were optimal for the monitoring of calcium concentration changes during neuronal activity, which typically varies from 50–100 nM at rest to 250–10,000 nM during activation in the cytosol of mammalian cells [35,36]. The high calcium affinity of the FGCaMP5 indicator is probably a result of the N15I substitution in the M13-peptide (Appendix A), which may strengthen its interactions with the Ca^2+^-binding domains of CaM. Indeed, N15I was a unique mutation among the FGCaMP5, FGCaMP6, and FGCaMP7 indicators; it has been previously shown that mutations in the M13-peptide remarkably affected the indicator’s affinity to calcium ions [19]. According to the Hill coefficient values calculated by fitting the equilibrium titration curves to the Hill equation, the FGCaMP5, FGCaMP6, and FGCaMP7 indicators bound calcium ions with slightly less co-operativity, compared to the control FGCaMP and GCaMP6s indicators (Hill coefficients: 1.6–2.3 vs. 2.3–4.0; Table 1 and Appendix A) [4,19]. Hill coefficients in the range of 3–4 are common for many GECIs, which implies linear dependence of the ΔF/F response to changes in calcium concentration in a narrow calcium ion concentration range but with higher ΔF/F response. When the Hill coefficient is close to 1 or less (which is characteristic of most synthetic dyes), the ΔF/F dependence vs. [Ca^2+^] is linear in a wider range of calcium ion concentrations [37,38]; however, this is at the expense of lower sensitivity. Hence, the enhanced versions of FGCaMP demonstrated advantageous mono-phasic fluorescence responses to calcium ions and had affinities to calcium ions appropriate for the monitoring of neuronal activity.

Next, we characterized another important property of the improved FGCaMPs GECIs: their pH stability. According to fluorescence changes, the 400 nm absorbing form of FGCaMP7 exhibited a slight shift in p*K*_a_ value (from 6.63 to 6.00) after the addition of calcium ions (Figure 1e and Table 1). Upon Ca^2+^ binding, the p*K*_a_ values for the 498 absorbing form shifted from 5.28 and 7.81 to 6.87 (Figure 1f and Table 1). In the range of pH values of 7–8, which is typical in the cytosol of neurons [39], the changes of fluorescence intensity for the protonated form in Ca^2+^-free state and anionic form in Ca^2+^-saturated state were 25% and 36%, respectively, while the change of the dynamic ranges for both forms in the pH range 7–8 was minor—which is an advantage of FGCaMP7 (Figure 1e,f) the protonated forms of FGCaMP5 and FGCaMP6 showed larger shifts in p*K*_a_s, from 7.01 to 5.75 and from 6.84 to 5.57, respectively (Appendix A), upon calcium-binding. The p*K*_a_ shifts of 496–497 nm absorbing forms for FGCaMP5 and FGCaMP6 were negligible (Appendix A), considering that their anionic forms are more pH-stable than the respective form of FGCaMP7. In contrast to the FGCaMP7 indicator, the dynamic ranges for the protonated forms of FGCaMP5 and FGCaMP6 changed by about 25%–30% in the range of pH values 7–8 (Appendix A). At the same time, changes in the dynamic ranges of anionic forms of FGCaMP5 and FGCaMP6 were negligible under the same conditions (Appendix A). Hence, the fluorescence response of these FGCaMP variants may be affected by variations in pH.

Next, we assessed the photobleaching stability for the FGCaMP7 indicator, as the indicator combining high contrast and brightness for both forms and showing the best performance in neuronal cultures (see results below). Both protonated and anionic forms of FGCaMP7 were extremely photostable in vitro. Under a wide-field microscope equipped with a metal halide lamp and 470/40BP excitation filter, FGCaMP7 in Ca^2+^-saturated state photobleached 2.7- and 1.8-fold slower than a control mEGFP protein and FGCaMP indicator in Ca^2+^-saturated state. In the absence of Ca^2+^ ions, FGCaMP7 photobleached 8.7- and 8.6-fold slower than mTagBFP2 and FGCaMP as a result of illumination with 355 ± 25 nm light, respectively (Figure 1g and Table 1). Hence, both 400- and 498-forms of the FGCaMP7 indicator demonstrated photostability in vitro higher than the photostabilities of FGCaMP, mTagBFP2, and mEGFP proteins.

Both in the absence of Ca^2+^ and the presence of 1 mM Ca^2+^, purified FGCaMP7 sensor eluted in size-exclusion chromatography as a monomer (Appendix A); note that monomeric proteins are usually preferable in terms of reduced cytotoxicity [40].

As fast Ca^2+^ association and dissociation kinetics are crucial for the resolution of neuronal dynamics, these kinetics were studied in the FGCaMP variants using stopped-flow fluorimetry. To investigate the association kinetics, we calculated the observed association rate constants (k_obs_) at [Ca^2+^] 300–1000 nM in the presence of 1 mM MgCl_2_ by fitting the kinetic curves of FGCaMP5, FGCaMP6, and FGCaMP7 to double exponentials at 400 nm excitation, to a single exponential at 498 nm excitation, and the kinetic curves of the original FGCaMP to a single exponential at both excitation wavelengths (Appendix A, and Figure 1h). The contribution of exponent 1 at 400 nm excitation varied from 10% to 29% for FGCaMP5, from 34% to 49% for FGCaMP6, and from 0% to 35% for FGCaMP7 (Appendix A and Figure 1i). According to the k_obs_ values obtained from the stopped-flow data, in the range of free Ca^2+^ concentrations 300–1000 nM, the FGCaMP5 indicator was the fastest among the FGCaMP mutants, which associated with calcium ions 1.1–1.9-fold faster than FGCaMP6 and FGCaMP7 at both 400 and 498 nm excitation (Appendix A). At 300 nM calcium concentration, as observed in neurons at one action potential (AP), FGCaMP5 associated with calcium ions 2.1/1.7-fold or 2.7/2.3-fold faster than control GCaMP6s or progenitor FGCaMP at 400/498 nm excitation, respectively. At the same time, at 300 nM [Ca^2+^] at 400 nm excitation, the FGCaMP7 indicator associated with calcium 2.1-fold slower than the fastest GCaMP6f indicator and 1.2- and 1.6-fold faster than GCaMP6s and FGCaMP (Figure 1h and Appendix A). At 498 nm excitation, FGCaMP7 associated with calcium ions was 3.0-fold slower than GCaMP6f and had an association rate constant comparable to the respective constants for GCaMP6s and FGCaMP (Figure 1h and Appendix A).

The dissociation half-time (t_1/2_^off^) of FGCaMP5 was 1.5–2.1-fold longer than the respective half-times for FGCaMP6 and FGCaMP7 (Appendix A). The FGCaMP7 indicator at 400 nm excitation released calcium ions 1.5- and 1.3-fold slower than GCaMP6s and FGCaMP and 4.1-fold slower than GCaMP6f, respectively (Figure 1j and Table 1). At 498 nm excitation, the dissociation half-time of FGCaMP7 was similar to the dissociation half-time of FGCaMP and 1.3- or 3.6-fold longer than those obtained with GCaMP6s or GCaMP6f, respectively (Figure 1j and Table 1). Hence, enhanced FGCaMPs have fast association–dissociation kinetics for calcium ions, comparable to the kinetics of the standard GCaMP6s GECI, which demonstrates their suitability for the visualization of fast neuronal activity.

Thus, using directed molecular evolution, we substantially improved the new class of FGCaMP calcium indicators, based on Ca^2+^-binding parts from fungi. As a result, for further characterization in HeLa cells and neuronal cultures, we selected three enhanced FGCaMPs variants which demonstrated a mono-phasic response to calcium ions, had different calcium affinities and dynamic ranges superior to that of FGCaMP, and which preserved the ratiometric phenotype of the original FGCaMP indicator.

### 2.4. Calcium-Dependent Response of Improved FGCaMP Indicators in HeLa Cells

To evaluate the behaviors of the enhanced FGCaMP indicators in live mammalian cells, we expressed FGCaMP5, FGCaMP6, and FGCaMP7, as well as control FGCaMP and GCaMP6s, in HeLa Kyoto cells together with a red reference R-GECO1 calcium indicator, and studied their response to ionomycin-induced elevation of calcium concentration in the cell cytosol [3]. After the addition of 2.5 μM ionomycin, we observed increases in red and green fluorescence when excited at 561 and 488 nm, respectively; green fluorescence at 405 nm excitation decreased (Figure 2 and Appendix A). The ΔF/F_0_ value for the fast calcium-associated FGCaMP5 normalized to ΔF/F_0_ of the R-GECO1 indicator was only 87 ± 25% at 488 nm excitation (Table 2). The mean normalized ΔF/F_0_ response for FGCaMP7 was 237 ± 81% from the R-GECO1 response, which was 2.1- and 1.5-fold higher than normalized responses for GCaMP6s (*p* ˂ 0.0001) and FGCaMP (*p* = 0.0045), respectively. The ΔF/F_0_ response for FGCaMP6 was similar (*p* = 0.9313) to that of the original FGCaMP and 1.4-fold higher (*p* = 0.0297) than the response of the control GCaMP6s indicator. At 405 nm excitation, the ΔF/F_0_ values for all three indicators (FGCaMP5, FGCaMP6, and FGCaMP7) were similar and 1.4–1.6-fold less than that of FGCaMP (*p* = 0.0243, *p* = 0.0230, and *p* = 0.0493, respectively; Table 2). Hence, the FGCaMP7 indicator demonstrated the highest ionomycin-induced ΔF/F_0_ response in cultured mammalian cells among the three FGCaMP variants tested and GCaMP6s, while FGCaMP5 had the lowest ΔF/F_0_ response; this correlated with its lowest contrast and highest calcium affinity in vitro (Table 2).

### 2.5. Visualization of Spontaneous Activity in Neuronal Cultures Using Enhanced Versions of FGCaMP Indicator

We next compared the enhanced FGCaMP variants in their ability to monitor calcium dynamics during the spontaneous activity of live neurons. We co-transduced dissociated neuronal cultures isolated from mice pups with recombinant adeno-associated viral particles (rAAV) carrying FGCaMP5, FGCaMP6, FGCaMP7, FGCaMP, and GCaMP6s GECIs together with control red R-GECO1. We observed calcium oscillations in all cultures by day 11–17 in vitro (Figure 3 and Appendix A). In the case of the ratiometric FGCaMP series, calcium oscillations were detected with opposite fluorescence changes at 405 and 488 nm excitation, respectively (Figure 3 and Appendix A). The mean ΔF/F_0_ response (normalized to ΔF/F_0_ of R-GECO1) was similar for FGCaMP, FGCaMP6, and FGCaMP7 at 405 nm excitation. However, ΔF/F_0_ for FGCaMP5 was 2.6-fold lower (*p* = 0.0014) than that of FGCaMP, perhaps because the protein was partially bound to Ca^2+^ in resting neurons due to its lower K_d_ value of 98 nM, as compared to the K_d_ value of 460 nM for FGCaMP. At 488 nm excitation, the mean normalized ΔF/F_0_ responses for FGCaMP7 and FGCaMP6 were comparable to that of FGCaMP (*p* = 0.8026 and *p* = 0.8895, respectively) and GCaMP6s (*p* = 0.4966 and *p* = 0.3932, respectively; Table 2). The mean normalized ΔF/F_0_ for FGCaMP5 at 488 nm excitation was 2.3–2.6-fold lower (*p* < 0.0001) than ΔF/F_0_ for FGCaMP, FGCaMP6, FGCaMP7, and GCaMP6s. Finally, we estimated ratiometric ΔR/R_0_ responses for the FGCaMP5, FGCaMP6, and FGCaMP7 indicators normalized to ΔF/F_0_ of R-GECO1 and compared them to ΔR/R_0_ for the original FGCaMP and ΔF/F_0_ for GCaMP6s. The mean normalized ΔR/R_0_ for FGCaMP7 was 1.6-fold higher (*p* = 0.0014) than ΔF/F_0_ of GCaMP6s, comparable to ΔR/R_0_ of FGCaMP (*p* = 0.3054) and FGCaMP6 (*p* = 0.6803), and 2.1-fold higher (*p* = 0.0002) than ΔR/R_0_ of FGCaMP5 (Table 2). Hence, during the spontaneous activity of neuronal cultures, FGCaMP7 revealed ΔF/F_0_ or ΔR/R_0_ responses similar or superior to those for other FGCaMP variants, FGCaMP, and GCaMP6s.

When comparing mean rise half-times at 488 nm excitation, the rise of FGCaMP7 fluorescence was slightly faster (1.36 ± 0.47 s) than the rise of the FGCaMP fluorescence (2.08 ± 0.57 s) and the other FGCaMP variants (FGCaMP5: 2.01 ± 1.23 s; FGCaMP6: 1.65 ± 0.71 s) and slightly slower than the rise of GCaMP6s fluorescence (1.14 ± 0.61 s). FGCaMP5 was the slowest indicator, with mean decay half-time values of 6.51 ± 2.50 and 5.78 ± 2.56 s at 488 and 405 nm excitation, respectively. Decay half-times for the FGCaMP7 (3.84 ± 0.88 and 3.40 ± 0.72 s at 488 and 405 nm excitation, respectively), FGCaMP (4.01 ± 1.65 and 3.56 ± 1.37 s at 488 and 405 nm excitation, respectively), and GCaMP6s (3.77 ± 1.27 s) GECIs were similar. Mean rise and decay half-times for the control co-expressed R-GECO1 indicator were similar to the respective half-times of FGCaMP7 (t_1/2_^rise^ R-GECO1 = 1.73 ± 0.63 s, t_1/2_^decay^ R-GECO1 = 4.17 ± 1.45 s).

Both FGCaMP7 and GCaMP6s indicators demonstrated even distribution in the cytosol of neurons, however R-GECO1 indicator revealed puncta-like aggregates during prolonged expression in neuronal cultures (Appendix A). This difference may be attributed to different fluorescent domains utilized in FGCaMP7 and GCaMP6s (EGFP fluorescent domain) as compared to R-GECO1 (mApple fluorescent domain). Puncta formation was attributed to lysosomes [41], therefore autophagy should adjust the number of these puncta [42].

Hence, all three enhanced FGCaMP variants were able to visualize spontaneous calcium activity in neuronal culture. According to the rise and decay half-times values—as well as to the ΔF/F_0_ and ΔR/R_0_ response values—during spontaneous neuronal activity, FGCaMP7 was chosen as the best variant among them. Its normalized ratiometric ΔR/R_0_ response was 1.6-fold higher (*p* = 0.0014) than the normalized ΔF/F_0_ response for GCaMP6s.

### 2.6. Responses of Improved FGCaMP Indicators to External Electric Stimulation of Neuronal Cultures

We next compared the responses of the enhanced FGCaMP variants, control FGCaMP, GCaMP6s, and R-GECO1 GECIs during external electric field stimulation of neuronal cultures. FGCaMP variants were co-expressed in neuronal cultures together with reference red R-GECO1 [3]. FGCaMP and GCaMP6s GECIs were used as controls. The number of APs induced by 150–300 external electric field pulses were evaluated according to the ΔF/F_0_ changes of co-expressed R-GECO1, assuming linearity of its response and 4% ΔF/F_0_ value at 1 AP stimuli [43]. 

All indicators responded to trains of 1–40 APs; except for FGCaMP and FGCaMP5, which started to sense trains from 10 APs and up (Figure 4a,b) and FGCaMP6, which responded to trains starting from 4 APs and up (Figure 4c). The green fluorescence in GCaMP6s-expressing neurons increased in response to electric field stimulation, while green fluorescence in neurons expressing ratiometric indicators FGCaMP, FGCaMP5, FGCaMP6, and FGCaMP7 increased at 488 nm excitation and decreased at 405 nm excitation. To compare the responses of enhanced FGCaMP variants, we estimated ΔF/F_0_ or ΔR/R_0_ values per 1AP, calculated as a slope of linear dependence of ΔF/F_0_ or ΔR/R_0_ vs. trains of 1–40 APs in the case of FGCaMP7 and GCaMP6s, 4–40 APs in the case of FGCaMP6, and 10–40 APs in the case of FGCaMP5 and FGCaMP (Figure 4). The ΔF/F_0_ response per 1 AP for FGCaMP7 at 405 nm excitation was 1.9–3.5-fold higher (*p* < 0.0001) than responses of FGCaMP5, FGCaMP6, and control FGCaMP (Table 2). At 488 nm excitation, ΔF/F_0_ response per 1AP for FGCaMP7 was 3.7- (*p* < 0.0001), 4.3- (*p* < 0.0001), and 2.1-fold (*p* < 0.0001) higher than the responses of FGCaMP, FGCaMP5, and FGCaMP6, respectively (Table 2). At 488 nm excitation, ΔF/F_0_ response per 1 AP for FGCaMP7 was similar (*p* = 0.8805) to that of GCaMP6s (Table 2). We also compared ratiometric ΔR/R_0_ responses per 1AP for enhanced FGCaMP variants with the respective ΔF/F_0_ value for GCaMP6s (Figure 4f and Table 2). The ΔR/R_0_ response per 1 AP for the FGCaMP7 variant with highest contrast in vitro was 3.8- (*p* < 0.0001), 4.0- (*p* < 0.0001), and 2.2-fold (*p* < 0.0001) higher than respective values for FGCaMP, FGCaMP5, and FGCaMP6, respectively. The ΔR/R_0_ response per 1 AP for FGCaMP7 was 1.4-fold higher (*p* = 0.0002) than the respective ΔF/F_0_ response per 1 AP for GCaMP6s (Table 2).

Thus, according to the external electric field stimulation of neuronal cultures, the enhanced FGCaMP variants, similarly to control FGCaMP and GCaMP6s GECIs, demonstrated linear dependence of ΔF/F response in the range of 0–40 APs. The FGCaMP7 indicator outperformed the FGCaMP progenitor and FGCaMP5/FGCaMP6 indicators, in terms of larger ΔF/F_0_ and ΔR/R_0_ responses. The ΔF/F_0_ response per 1 AP for FGCaMP7 at 488 nm excitation was similar to that of the GCaMP6s control, while ratiometric ΔR/R_0_ response per 1 AP for FGCaMP7 was 1.4-fold higher (*p* = 0.0002) than ΔF/F_0_ per 1 AP for GCaMP6s. The FGCaMP7 variant was considered as best and chosen for further characterization in vitro, in mammalian cells, and in vivo.

### 2.7. Characterization of Truncated Versions (with Deleted M13-Like Peptide) of FGCaMP7 and GCaMP6s Indicators In Vitro and in HeLa Cells

To suggest the mechanism of fluorescence changes of the FGCaMP7 and GCaMP6s calcium indicators caused by calcium ion binding and to assess the possible interactions between these indicators and the intracellular environment, we constructed truncated versions of FGCaMP7 and GCaMP6s with deleted M13-like peptide (called FGCaM7 and GCaM6s, respectively) and studied their responses to calcium ions in vitro as purified proteins and in the cytosol of mammalian cells. These truncated versions, as purified proteins, practically did not respond to the addition of calcium ions in the range of 39–2000 µM free-calcium ion concentrations (Appendix A). Hence, the CaM domain binding to calcium ions does not evoke fluorescent changes in the FGCaMP7 and GCaMP6s indicators, per se. It is likely that only subsequent M13-like peptide binding with the calcium-saturated CaM domain is translated into the fluorescence response of these indicators.

To assess the possible interactions of the FGCaMP7 and GCaMP6s proteins with the intracellular environment, we analyzed the responses of the FGCaM7 and GCaM6s truncated versions to ionomycin-induced calcium ion elevations in HeLa cells. The FGCaM7 and GCaM6s truncated versions transiently expressed in the HeLa cells had an even distribution in the cytosol of the cells (Figure 5a,b, respectively). FGCaM7 practically did not respond to the elevation of calcium ions induced by ionomycin addition, with ΔF/F_0_ values of 6.5 ± 2.4% (Figure 5a,c,d). However, GCaM6s sensed the ionomycin-induced elevation of calcium ions, with an ΔF/F_0_ response of 139 ± 62% (Figure 5b–d). Although this response was 4.3-fold lower (*p* < 0.0001), as compared to the ΔF/F_0_ response for the full-length GCaMP6s of 600 ± 333% under the same conditions, it was 21-fold larger (*p* < 0.0001) than that of FGCaM7. Taking into account that purified FGCaM7 and GCaM6s proteins practically did not respond to saturating calcium ion concentrations, we concluded that the CaM part of the GCaM6s protein interacted with intracellular proteins, as opposed to that in the FGCaM7 indicator. Earlier, the fluorescence recovery after photobleaching (FRAP) experiments demonstrated that the progenitor of the FGCaMP7 indicator, FGCaMP, was freely diffusible in the cytoplasm of HeLa cells at physiological Ca^2+^ concentrations, but that about 30% of GCaMP6s and G-GECO1.2 were not freely diffusible, suggesting that they may become bound to CaM or other cellular proteins [19]. Therefore, the GCaM6s protein is prone to interaction with intracellular proteins, in contrast to the FGCaM7 protein. We speculated that this difference may be attributed to the fungal origin of the CaM part in the FGCaM7 indicator, which has a different amino acid sequence than metazoan CaM. Overall, the M13-like peptide complex formation with calcium-bound CaM, rather than calcium ion binding by CaM domain, was translated into the fluorescence response of the GCaMP6s and FGCaMP7 indicators; in mammalian cells, the CaM domain of the FGCaMP7 indicator practically did not interact with intracellular proteins, in contrast to the CaM domain of the GCaMP6s indicator. 

### 2.8. In Vivo Imaging of Hippocampal Neuronal Activity in Freely Moving Mice Using NVista Miniscope and FGCaMP7 Calcium Indicator

To validate the in vivo application of the FGCaMP7 indicator using one-photon non-ratiometric imaging, we performed in vivo calcium imaging with an NVista head-mounted microscope (Figure 6a) and compared the amplitudes and kinetics of spikes detected during specific and non-specific activities of neurons in the CA1 field of the hippocampus while mice explored an O-shaped track with landmarks. The FGCaMP7 indicator was delivered into CA1 hippocampal neurons of 1–2 month-old mice using rAAVs carrying NES-FGCaMP under CAG promoter. Several weeks after lens probe implantation, we recorded neuronal activity in the hippocampus of the mice while they explored an O-shaped track with landmarks. Using the MIN1PIPE procedure pipeline [44] and manual inspection, we successfully identified active cells and extracted respective calcium activity ΔF/F_0_ traces. An example of typical neuronal activity for eight neurons is shown in Figure 6b. Averaged peak ΔF/F_0_ for FGCaMP7 was 1.0 ± 1.2, which was 2.1-fold lower (*p* < 0.0001) than averaged ΔF/F_0_ for GCaMP6s (Figure 6c and Appendix A). Averaged rise and decay half-times for spikes detected for FGCaMP7 (0.97 ± 0.60 and 3.07 ± 0.80 s, respectively) were 1.3- (*p* = 0.0002) and 1.2-fold (*p* < 0.0001) longer than the respective times of the GCaMP6s indicator (0.76 ± 0.49 and 2.54 ± 0.94 s, respectively; Appendix A). Therefore, the FGCaMP7 indicator detected hippocampal neuronal activity with slightly less sensitivity and resolution than the control GCaMP6s.

We could not reveal a noticeable difference between the FGCaMP7 and GCaMP6s indicators according to the appearance of cells having an unusual shape and intracellular aggregates even upon prolonged in vivo expression of these indicators in hippocampus of mice (Appendix A). A separate study with better microscopic resolution and more statistics is needed to identify possible difference in cellular toxicity between FGCaMP7 and GCaMP6s.

We next compared the ability of the FGCaMP7 and GCaMP6s indicators to visualize the location-specific activity of CA1 cells using an nVista miniscope [8]. We correlated the neuronal calcium activity in the CA1 area of the hippocampus while the mouse moved through an O-shaped track with landmarks (Figure 6a). As a result, using both indicators, we identified the neurons that were specifically activated in certain parts of the track (an example of an FGCaMP7-labeled place cell is shown in Figure 6d, as well as Videos S1 and S2). The ΔF/F responses of FGCaMP7 (1.0 ± 0.6) averaged across the space-specific activity of neuronal place cells were 3.7-fold lower (*p* < 0.0001) than the respective responses of the GCaMP6s indicator (3.7 ± 0.9) (Figure 6e). Thus, FGCaMP7 detected both non-specific and specific neuronal activity with an ΔF/F response 2.1–3.7-fold lower than GCaMP6s. 

### 2.9. In Vivo Imaging of Hippocampal Neuronal Ensembles in Mice during Food Intake Using NVista Miniscope and FGCaMP7 Calcium Indicator

To demonstrate the capability of the FGCaMP7 indicator to reveal neuronal ensembles, we performed in vivo calcium imaging of the mouse hippocampus using an NVista head-mounted microscope during food intake sessions. The FGCaMP7 indicator was delivered into CA1 hippocampal neurons using rAAVs carrying NES-FGCaMP under CAG promoter. Ca^2+^ imaging was performed for 1 h in the home cage with two weighed cups of familiar powdered food after food deprivation for 24 h (Figure 7a). Based on a calcium activity ΔF/F_0_ trace extraction routine described earlier [6], we correlated neuronal activity of the CA1 hippocampus with the trajectory of the mouse (Figure 7b) and detected moments of transitions to the food cup area (Figure 7c). To analyze neuronal response dynamics during the transition to the food cup area, we averaged z-scored neuronal Ca^2+^ responses for each neuron in a time window from 2 s before the transition to 5 s after [45]. The mean responses of all neurons from two mice were clustered into three groups using k-means (Figure 7d), which revealed a generally activated ensemble (40% of neurons), a generally inhibited ensemble (19% of neurons), and an ensemble with neutral responses (Figure 7e). The mean z-scored ΔF/F_0_ response to cup area-entry transitions showed an increase by 0.9 (*p* < 0.0001) for the activated ensemble and a decrease by 0.3 (*p* < 0.0001) for the inhibited ensemble (Figure 7f). Similarly, three types of ensembles in the amygdala of mice were observed in other types of behavioral tasks [45,46]. The mean z-scored ΔF/F_0_ responses for these ensembles were the same, as compared to our data. Overall, using the FGCaMP7 indicator and population analysis of neuronal activity, we identified three types of neuronal ensembles in the hippocampus of mice during food intake.

### 2.10. Two-Photon In Vivo Imaging of Visual Cortex in Awake Mice Expressing FGCaMP7 Calcium Indicator

To estimate the two-photon performance of the novel FGCaMP7 indicator in vivo, we assessed its response to spontaneous neuronal activity in awake mice under a two-photon microscope. rAAV particles of the FGCaMP7 indicator with NES signal were injected into the brain of mice pups at the age of P0–P1 using a Hamilton syringe. A month after viral infection, a cranial window was implanted and, one month later, two-photon imaging of spontaneous neuronal activity was performed in the L2/3 layer of the visual cortex in awake mice. FGCaMP7-expressing neurons were identified using both 800 and 960 nm excitation light, which excited protonated and anionic forms of the indicator, respectively. Both forms of FGCaMP7 demonstrated cytosolic nuclei-excluded expression (Figure 8a,b; left). 

Spontaneous neuronal activity was observed at a depth of 40–250 μm. An example of neuronal activity traces at 960 nm excitation is demonstrated in Figure 8a (right). Though we observed green fluorescence in FGCaMP7-expressing neurons at 800 nm excitation, we could not detect any fluorescence changes associated with Ca^2+^ dynamics in neurons using this excitation light; perhaps due to the 3.5-fold lower ΔF/F_0_ response for the 400 nm absorbing form, as compared to the response for the 498 nm absorbing form in vitro (Figure 8b, right; Appendix A; and Table 1). Two-photon excitation of the blue-shifted absorption peak of the NIR-GECO1 indicator also did not result in the observation of fluorescence changes associated with neuronal calcium dynamics in cultured neurons and in vivo [47]. At 960 nm excitation, we observed neuronal activity-dependent changes in fluorescence of the FGCaMP7 anionic form, with a peak ΔF/F_0_ of 0.37 ± 0.23 (Figure 8c). In the case of GCaMP6s, peak ΔF/F_0_ was 0.45 ± 0.28 (Figure 8d), which was 1.2-fold higher (*p* = 0.0001) than that of FGCaMP7 (Figure 8e). Signal-to-noise ratios (SNR) were the same (*p* = 0.6979) for FGCaMP7 (6.22 ± 4.32) and GCaMP6s (5.82 ± 3.70) (Figure 8e). FGCaMP7 was 1.4–1.5-fold (*p* < 0.0001) faster, in terms of rise (1.06 ± 0.79 s) and decay (2.07 ± 1.09 s) half-times, when compared with the rise (1.58 ± 0.88 s) and decay (2.88 ± 1.16 s) half-times of GCaMP6s. Hence, the FGCaMP7 indicator could visualize spontaneous neuronal activity in the visual cortex of mouse brain in vivo using two-photon excitation of its anionic form with 960 nm light; in contrast to its second protonated form (excited with 800 nm two-photon light), which was fluorescent but did not respond to neuronal activity.

### 2.11. In Vivo Ratiometric Two-Color Imaging of Neuronal Activity in Zebrafish Using FGCaMP7

To test whether FGCaMP7 can report neuronal activity in vivo under one-photon ratiometric imaging, we transiently expressed NES-FGCaMP7 under the pan-neuronal promoter *elavl3* in zebrafish larvae and imaged spontaneous activity (see Methods). Both 400- and 498-forms of the FGCaMP7 indicator were observed using 405 and 488 nm excitation light, respectively (Figure 9a). The FGCaMP7 indicator was excluded from nuclei and observed in the soma and processes of neurons. We performed live-imaging to record spontaneous activity in nine neurons from six larvae. Fluorescence transients were observed for both forms (Figure 9b), except for the 400-form in one cell. We calculated the average peak ΔF/F_0_ value and rise/decay half-times for both forms across all imaged cells using the averaged calcium transient for each cell (Figure 9c,d). The average peak ΔF/F_0_ value at 488 nm excitation was 1.64 ± 0.87 (mean ± SEM), which was about 9.6-fold higher than the ΔF/F_0_ value of −0.17 ± 0.05 at 405 nm excitation. The average ΔR/R_0_ value for the ratio of the observed signals (488 nm signal over 405 nm signal) was 3.13 ± 1.98 (Figure 9d). The average rise and decay half-times were similar for both forms: t_1/2_^on^ 1.47 ± 0.31 s for 488 nm and 1.45 ± 0.29 s for 405 nm; while t_1/2_^off^ 6.25 ± 1.37 s for 488 nm and 7.19 ± 2.30 for 405 nm (Figure 9d). These results demonstrate that the FGCaMP7 indicator can be used to ratiometrically monitor calcium transients in vivo in larval zebrafish with a high signal-to-noise ratio. 

## 3. Materials and Methods

### 3.1. Protein Purification for X-ray Crystallography

The bacterial cells expressing the FCaMP7 protein with a His-tag and Tobacco Etch Virus (TEV) protease cleavage site were harvested by centrifugation, resuspended in 40 mM Tris-HCl buffer, pH 7.8 supplemented with 400 mM NaCl, 10 mM Imidazole, 0.2% Triton X-100 and 1 mM PMSF and disrupted by sonication. The crude cell extract was centrifuged for 30 min at 28,000 *g* and 4 °C. The supernatant was applied to a Ni-NTA Superflow column (Qiagen, Hilden, Germany) equilibrated with the binding buffer (40 mM Tris-HCl, pH 7.8, containing 400 mM NaCl, 10 mM imidazole and 0.1% (*v*/*v*) Triton X-100). Washing and elution were performed with the same binding buffer without Triton X-100 and supplemented with 40 mM and 300 mM imidazole, respectively. Eluted protein was concentrated using a 10 kDa cutoff centrifugal filter device (Millipore, Burlington, MA, USA), and transferred into 40 mM Tris-HCl buffer, pH 7.8, containing 200 mM NaCl, 1 mM EDTA, 1 mM β-mercaptoethanol, 5 mM Imidazole and TEV protease (1 mg per 10 mg of protein). The solution was incubated overnight at +4 °C, dialyzed against the binding buffer and applied to a Ni-NTA Superflow column (Qiagen, Hilden, Germany). TEV protease and cleaved His-tag were absorbed onto the column Ni-NTA Superflow column (Qiagen, Hilden, Germany) and the flow-through was concentrated, exchanged to 20 mM TrisHCl buffer pH 7.8 supplemented with 5 mM CaCl_2,_ and applied to a ResourceQ column (GE Healthcare, Chicago, IL, USA) equilibrated with the same buffer. The column was washed with a linear gradient of NaCl from 0 to 1M. The major peak containing recombinant FCaMP7 was detected at NaCl concentration about 26%, its fractions were pooled, concentrated, and stored at −70 °C.

### 3.2. Crystallization of FGCaMP

An initial crystallization screening of FGCaMP was performed with a robotic crystallization system (Rigaku, Tokyo, Japan) and commercially available 96-well crystallization screens (Hampton Research and Anatrace, Aliso Viejo, CA, USA) at 15 °C using the sitting drop vapor diffusion method. The protein concentration was 7 mg/mL in the following buffer: 20 mM Tris-HCl, 300 mM NaCl, pH 7.5 supplemented with 5 mM CaCl_2_. Optimization of the initial conditions was performed by the hanging-drop vapor-diffusion method in 24-well VDX plates. Rod-like crystals were obtained within 2 weeks in the following conditions: 2.15M Ammonium sulfate, 10% Dimethyl sulfoxide.

### 3.3. Data Collection, Processing, Structure Solution and Refinement

FGCaMP crystals were briefly soaked in a 20% DMSO immediately prior to diffraction data collection and flash-frozen in liquid nitrogen. The X-ray data were collected from a single crystal at 100 K at the beamline BL41XU of the Spring8 synchrotron (Japan Synchrotron Radiation Research Institute, Hyōgo Prefecture, Japan). The data were indexed, integrated, and scaled with iMosflm [48] (Appendix A). Based on the L-test [49] the dataset was not twinned. The program Pointless [50] suggested the P4_3_2_1_2 space group.

Structure was solved by the molecular replacement method using the MOLREP program [51] and the structure of the GCaMP2 calcium sensor (PDB ID 3EK7) as an initial model. The refinement of the structure was carried out using the REFMAC5 program of the CCP4 suite [52]. Translation Libration Screw-motion refinement (TLS) was introduced at the earlier stages of refinement. Map sharpening was performed to improve its quality as well as non-crystallographic symmetry (NCS). The visual inspection of electron density maps and the manual rebuilding of the model were carried out using the COOT interactive graphics program [53]. The resolution was successively increased to 3.18 Å. In the final model, an asymmetric unit contained three independent copies of the protein (subunits A, B, and E) with one of them (E) being only a partial structure of CaM. The loss of a major part of the structure of the E subunit could be explained by limited proteolysis, which happened during the long crystallization process. The other two subunits consisted of 412 residues, of which regions 102–121 (101–121 and 342–347 for subunit B) and three residues from the C-terminal part have no electron density possibly due to high flexibility. Both A and B subunits contained chromophore together 8 calcium ions, and E subunit contained 2 calcium ions only.

### 3.4. Structure Analysis and Validation

The visual inspection of the structure was carried out using the COOT program and the PyMOL Molecular Graphics System, Version 1.9.0.0 (Schrödinger, LLC, New York, NY, USA). The structure comparison and superposition were made using the PDBeFold program [54], while contacts were analyzed using the PDBePISA [55] and WHATIF software [56]. The chromophore environment was visualized by LigPlot [57].

### 3.5. Mutagenesis and Library Screening

Mutagenesis and screening of bacterial libraries was performed sequentially on Petri dishes and purified proteins as described in reference [8]. Primers used are listed in Appendix A.

### 3.6. Proteins Purification and Characterization

Proteins were expressed, purified on a low-scale format, and characterized as described in references [8,19].

To determine the oligomeric state of the FGCaMP7, the protein (1.7 mg/mL concentration) was applied to a Superdex 200 10/30 GL column (GE Healthcare, Chicago, IL, USA) equilibrated with 20 mM Tris-HCl, pH 7.5, 1 mM CaCl_2_, or 2 mM EDTA.

### 3.7. Stopped-Flow Fluorimetry

The kinetic curves of Ca^2+^-association with FGCaMP and its variants were acquired on a Chirascan Spectrofluorimeter equipped with a stopped-flow module (Applied Photophysics, Leatherhead, Surrey, UK). Ca^2+^ (300 nM, 700 nM, and 1000 nM) and protein solutions (20 μg/mL in 30 mM HEPES buffer (pH 7.2) containing 100 mM KCl and 1 mM MgCl_2_) were prepared as described in [6]. Fluorescence excitation was set to 498 or 400 nm for FGCaMPs. Fluorescence emission was detected using a 515 nm cut-off filter. Exponential fitting of the fluorescence signal changes over time and fitting the observed data to the equation k_obs_ = k_on_ × [Ca^2+^]^n^ + k_off_ were performed as described in [6]. In the Ca^2+^- dissociation kinetics experiments, protein solutions (20 μg/mL) in 30 mM HEPES (pH 7.2), 100 mM KCl, 1 mM MgCl_2_, and 1 µM CaCl_2_ was rapidly mixed (1:1) with 30 mM HEPES (pH 7.2), 100 mM KCl, 1 mM MgCl_2_, and 10 mM EGTA.

### 3.8. Mammalian Plasmid Construction

To construct the pAAV-CAG-NES-mCherry plasmid, the NES-mCherry gene was PCR amplified as the BamHI-BsrGI fragment and swapped with the iRFP-P2A-EGFP gene in the pAAV-CAG-iRFP-P2A-EGFP vector. To construct pAAV-CAG-NES-FGCaMP5, pAAV-CAG-NES-FGCaMP6, pAAV-CAG-NES-FGCaMP7, and pAAV-CAG-NES-R-GECO1 plasmids, FGCaMP5, FGCaMP6, FGCaMP7, and R-GECO1 were PCR amplified as the BglII-HindII and BglII-EcoRI fragments, respectively and swapped with the mCherry gene in the pAAV-CAG-NES-mCherry vector.

### 3.9. Cell Culture and Transfection

HeLa Kyoto cells were cultured and transfected as described in reference [7].

### 3.10. Mammalian Live-Cell Imaging

HeLa Kyoto cell cultures were imaged 24–48 h after transfection using a laser spinning-disk Andor XDi Technology Revolution multi-point confocal system (Andor Technology, Belfast, UK) equipped with an inverted Nikon Eclipse Ti-E/B microscope (Nikon Instruments, Melville, NY, USA), a 75 W mercury-xenon lamp (Hamamatsu, Iwata City, Japan), a 60× oil immersion objective NA 1.4 (Nikon Instruments, Melville, NY, USA), a 16-bit Neo sCMOS camera (Andor Technology, Belfast, UK), laser module Revolution 600 (Andor Technology, Belfast, UK), spinning-disk module Yokogawa CSU-W1 (Andor Technology, Belfast, UK), and a cage incubator (Okolab, Pozzuoli, Italy). The green and red fluorescence were acquired using 80% of the 488 nm (17.3 µW/cm^2^ before objective lens) and 80% of 561 nm (62.3 µW/cm^2^ before objective lens) laser powers, confocal dichroic mirror 405/488/561/640 and filter wheel emission filters 525/50 and 617/73, respectively. 

For time-lapse imaging experiments with varying Ca^2+^ concentration, 2.5 μM ionomycin was added to cells for imaging calcium indicators in the Ca^2+^-saturated state.

### 3.11. rAAV Particles Production and Isolation

The rAAV particles were purified as described previously [7].

### 3.12. Isolation, Transduction, and Imaging of Neuronal Cultures

Dissociated neuronal cultures were isolated from C57BL/6 mice at postnatal days 0–3 and were grown on 35-mm MatTek glass-bottom dishes in Neurobasal Medium A (GIBCO, UK) supplemented with 2% B27 Supplement (GIBCO, UK), 0.5 mM glutamine (GIBCO, UK), 50 U/mL penicillin, and 50 μg/mL streptomycin (GIBCO, UK). On the 4th day in vitro, neuronal cells were transduced with 1–2 μL rAAV viral particles carrying AAV-*CAG*-NES-FGCaMP4 and AAV-*CAG*-NES-R-GECO1. Cells were imaged using an Andor XDi Technology Revolution multi-point confocal system as described above. ΔR/R_0_ was measured as (I_488_(Ca^2+^)/I_405_(Ca^2+^) − I_488_(noCa^2+^)/I_405_(noCa^2+^))/I_488_(noCa^2+^)/I_405_(noCa^2+^), where I_488_(Ca^2+^) and I_405_(Ca^2+^) are intensities of the indicator obtained under Ca^2+^-saturated conditions during spontaneous activity or in response to electric stimulation at 488 or 405 nm excitation, while I_488_(noCa^2+^) and I_405_(noCa^2+^) are intensities obtained under Ca^2+^ depletion at the same conditions.

### 3.13. Stimulation of Neuronal Cultures with Electric Field

The stimulation of neuronal cultures with the electric field were performed according to the described protocol [8]. Briefly, transduced neurons co-expressing the FGCaMP variant (FGCaMP, FGCaMP5, FGCaMP6, or FGCaMP7) or GCaMP6s with reference R-GECO1 were stimulated in the full Neurobasal medium supplemented with a 10 μM 6-Cyano-2,3-dihydroxy-7-nitro-quinoxaline (CNQX) and 100 μM 2-Amino-5-phosphonovalerate (APV) to inhibit spontaneous activity. APs were evoked by field stimulation with a custom-built stimulation unit and a custom-built 35 mm cap stimulator with pairs of parallel platinum-iridium wires. Fluorescence changes were monitored using Andor XDi Technology Revolution multi-point confocal system. ΔR/R_0_ was measured as (I_488_(Ca^2+^)/I_405_(Ca^2+^) − I_488_(noCa^2+^)/I_405_(noCa^2+^))/I_488_(noCa^2+^)/I_405_(noCa^2+^), where I_488_(Ca^2+^) and I_405_(Ca^2+^) are intensities of the indicator obtained under Ca^2+^-saturated conditions during spontaneous activity or in response to electric stimulation at 488 or 405 nm excitation, while I_488_(noCa^2+^) and I_405_(noCa^2+^) are intensities obtained under Ca^2+^ depletion at the same conditions.

### 3.14. Surgery and Imaging with an nVista HD Miniature Microscope

Two adult male C57BL/6 mice, aged 20 weeks at the start of the experiments, were used for this study. Mice underwent two surgical procedures under zoletil-xylazine anesthesia (40 and 5 mg/kg, respectively). First, a circular 2-mm-diameter craniotomy was made, and 500 nl of rAAV viral particles (carrying AAV-*CAG*-NES-FGCaMP7) was injected through a 50 µm tip diameter glass micropipette (Wiretrol I, 5-000-1001, Drummond, Broomall, PA, USA) into two areas of the hippocampus: CA1 (left hemisphere; stereotaxic coordinates:−1.9 mm A/P from bregma, −1.4 mm M/L, −1.3 mm D/V) or DG (left hemisphere: stereotaxic coordinates: −2.0 mm A/P from bregma, −1.6 mm M/L, −2.0 D/V). Microinjections were performed using UltraMicroPump with Micro4 Controller (WPI Inc., Sarasota, IL, USA) at a rate of 100 nl/min. All exposed surfaces of the brain tissue were sealed with KWIK-SIL silicone adhesive (WPI Inc., Sarasota, FL, USA). Two weeks later, the silicone was removed, and the dura matter was extracted from the craniotomy site. Then, a GLP 1040 (for CA1) and GLP 0561 (for DG) lens probes (Inscopix Inc., Palo Alto, CA, USA) was lowered slowly to a depth of 1.1 mm (for CA1) or 1.8 mm (for DG) while constantly washing the craniotomy site with sterile cortex buffer. Next, all the exposed tissue was sealed with KWIK-SIL and white dental cement (Stoelting, Wood Dale, IL, USA). Imaging data analysis was performed with custom MATLAB scripts [6] or MATLAB scripts, based on NoRMCorre [58] and MIN1PIPE [44] routines. First, imaging data were downsampled by factor 2 for increasing computation speed, and then displacements were corrected using the NoRMCorre routine. After that, neuron locations and traces were extracted using the MIN1PIPE pipeline. Then, calcium events exceeding 4 MADs were detected using approaches described earlier [8]. Video tracking was performed via open-source visual programming media Bonsai [59]. 

Place fields were estimated using a conservative approach: cell was considered spatial selective only if its activation was observed more than in 50% visits of putative place field zone. 

The ΔF/F_0_ value was calculated as (F(t)−F_0_)/F_0_, where F_0_ was the mean fluorescence value across all frames, F(t)—the fluorescence value at a time t. Z-score was calculated as (x(t)−$\bar{x}$)/SD, where x(t) was the ΔF/F_0_ value at time t, $\bar{x}$, and SD were the mean and standard deviation of the ΔF/F_0_ values over the entire individual neuronal trace, respectively. K-means clustering of neuron responses was based on minimization the function V =$\overset{k}{\underset{i=1}{\sum }}\underset{x\in S_{i}}{\sum }{\left(x-\mu_{i}\right)}^{2}$, where k was the number of clusters, $S_{i}$ were the resulting clusters, i = 1,2,…,k, $\mu_{i}$ were the centers of mass of all vectors x from the cluster $S_{i},$ x vectors were the mean z-scored response traces.

### 3.15. Viral Injection to the Neonatal Mouse Brain and Surgery for in Vivo Two-Photon Imaging

P0–P1 pups of C57BL/6 mice (Jackson Laboratory, Bar Harbor, ME, USA) were collected from the cage and prepared for viral injection by cryoanesthesia. The injection to the neonatal brain was performed in accordance with the following protocol [60,61]. Briefly, rAAV were diluted in PBS containing 0.05% Trypan blue (NanoEnTek, Seoul, South Korea) and a volume of 2.5–7.5 μL of the final solution was injected to the right hemisphere by 5 μL Hamilton syringe. The needle was held perpendicular to the skull surface to the depth of 1.5–2 mm. The pups were then placed on the warming pad for 3–5 min and transferred back to the mother cage. Surgery for in vivo two-photon imaging was performed 5–7 weeks post-infection as described in reference [8]. 

### 3.16. Two-Photon in Vivo Imaging in Mouse V1

Two-photon imaging of mouse brain was performed 60–80 days after viral injection and 30–45 days after cranial window implantation using an Olympus MPE1000 two-photon microscope equipped with a Mai Tai Ti:Sapphire femtosecond-pulse laser (Spectra-Physics, Santa Clara, CA, USA) and a water-immersion objective lens at 1.05 NA (Olympus, Waltham, MA, USA). Wavelengths of 800 and 960 nm were used for excitation. Images were acquired using the Olympus software. Functional images (256 × 256 pixels, 0.429 s per frame) of V1 neurons (40–250 μm under the pia) were recorded. Calcium activity traces were extracted from manually identified neuron contours for each acquired stack. Then ΔF/F_0_ normalization was performed for each trace separately and spike detection routine was performed as described earlier [7]. We searched single calcium events with an amplitude of not less than 3 MAD, fitted them with a 2-term exponential model and estimated their basic parameters: peak ΔF/F_0_, SNR, rise and decay time constants.

### 3.17. In Vivo Imaging of Neuronal Activity in Zebrafish Larvae

In vivo fluorescent imaging of FGCaMP7 was performed in 6–8 days post-fertilization (DPF) zebrafish larvae (Danio rerio). NES-FGCaMP7 fragment was PCR amplified from pAAV-CAG-NES-FGCaMP7 plasmid and cloned into the Tol2 transposon vector under *elavl3* promoter. The plasmid was injected with Tol2 mRNA into 1–2 cell stage embryos of the Tüpfel long-fin (TL) nacre strain. Larvae were paralyzed with α-Bungarotoxin (1 mg/mL) to prevent motion artifacts and embedded in 1.5% agarose gel in 3.5-cm Petri dishes. Spontaneous neuronal activity was recorded for 400- and 498-forms by switching between two excitation wavelengths of 405 nm and 488 nm and detecting emission with 493 LP filter. Images were acquired under Zeiss LSM700 confocal microscope with 40× water immersion objective (NA 1.0) at 420–550 ms per frame.

### 3.18. Analysis of FGCaMP7 Dynamics in Zebrafish Larvae

Drift in acquired images was corrected with ImageJ plugin TurboReg. Single spikes from a cell were selected manually and aligned to the peaks, and then the decay half-time was measured by fitting exponential to the averaged spike. The average ΔF/F_0_ was calculated as (F−F_0_)/F_0_, where F_0_ was set to the value of the most frequent bin in a 5-bin histogram of the baseline of averaged spikes. For the whole ΔF/F_0_ trace plot, F_0_ was set to the value of the most frequent bin in a 100-bin histogram of fluorescence intensity or the ratio trace.

### 3.19. Statistics

To estimate the significance of the difference between two values, we used the Mann–Whitney Rank Sum Test and provided *p*-values (throughout the text in the brackets) calculated for the two-tailed hypothesis. We considered difference as significant if the *p*-value was < 0.05.

### 3.20. Ethical Approval and Animal Care

All methods for animal care and all experimental protocols were approved by the National Research Center “Kurchatov Institute” Committee on Animal Care (NG-1/109PR of 13 February 2020) and were in accordance with the Russian Federation Order Requirements N 267 M3. All animal experimental procedures were approved by the Max Planck Society and the local government (Regierung von Oberbayern). Eleven C57BL/6 mice were used in this study, ages P0-P3 and ~2–3 months. Mice were used without regard to gender. Six zebrafish larvae at age 6–8 DPF were used in the study.

## 4. Conclusions

In conclusion, we resolved the X-ray structure of the previously developed ratiometric FGCaMP calcium indicator [19], based on M13-like peptide/CaM from *Aspergillus niger* and *Aspergillus fumigatus* fungi with amino acid sequences differing from the homologous sequences of Ca^2+^-binding parts of metazoans typically used in calcium indicators. Based on the crystal structure analysis, we found positions that stabilize the anionic form of FGCaMP and which potentially may serve as hotspots for further mutation of GFP-like FP-based GECIs. To address the limitations of the FGCaMP indicator, such as complex fluorescence response to calcium ions and limited dynamic range and sensitivity in neurons, we further developed novel FGCaMP variants—FGCaMP5, FGCaMP6, and FGCaMP7—using a random mutagenesis strategy. We characterized their main properties in vitro, in HeLa cultured cells, and neurons during spontaneous and induced activity. FGCaMP7 appeared to be the most promising variant among developed mutants. 

FGCaMP7 preserved the ratiometric phenotype of FGCaMP with a ratiometric contrast 3.0-fold higher than the contrast of FGCaMP. The main application of GECIs is in vivo photometry, and the advantage of the non-ratiometric GCaMP6 indicators in this regard is the ability to use 405 nm as the “isobestic” wavelength, where calcium does not affect fluorescence [62,63]. This is typically used to normalize signals and remove motion artifacts. In this type of application, the control 405 nm light should be avoided in the case of ratiometric FGCaMP7 and other control wavelengths such as 561 or 640 nm might be used.

FGCaMP7 demonstrated a mono-phasic calcium titration curve with a higher sensitivity to calcium ions than FGCaMP. 

Similarly to GCaMPs, FGCaMP7 has four calcium-binding sites per one molecule. Hence, its overexpression in cells may affect calcium ions concentration in the same way as GCaMPs [64]. Hence, FGCaMP7 does not have an advantage in the respect of calcium buffering.

Under stimulated activity in cultured neurons at 488 nm excitation, the averaged ΔF/F_0_ per 1 AP for FGCaMP7 was 3.7-fold higher (*p* < 0.0001) than ΔF/F_0_ per 1 AP for FGCaMP and similar (*p* = 0.8805) to that of GCaMP6s. Ratiometric ΔR/R_0_ response per 1 AP for FGCaMP7 was 3.8-fold higher (*p* < 0.0001) compared to the respective value for FGCaMP and 1.4-fold higher (*p* = 0.0002) than that of GCaMP6s. 

We also showed that a truncated variant of FGCaMP7 with deleted M13-like peptide (called FGCaM7) did not respond to ionomycin-induced elevation of calcium ions in HeLa cells, in contrast to the GCaM6s truncated variant, which responded to ionomycin addition with ΔF/F_0_ of 139 ± 62%. As both FGCaM7 and GCaM6s purified proteins did not respond to calcium ion addition in vitro, we suppose that GCaMP6s (based on mammalian Ca^2+^-binding parts) is prone to interactions with the intracellular environment; however, FGCaMP7 and the earlier-developed FGCaMP (both based on fungal Ca^2+^-binding domains) are less prone to such interactions. 

We demonstrated that FGCaMP7 is suitable for application in monitoring in vivo neuronal activity in mouse hippocampus using an nVista miniscope. Ratiometric calcium imaging of the FGCaMP7 indicator using two-photon microscopy in vivo in mouse cortex was hampered by the absence of any response of the protonated form with 800 nm excitation to neuronal activity; however, the anionic form with 960 nm excitation did respond to neuronal activity. Finally, we demonstrated ratiometric calcium imaging of neuronal activity in zebrafish using confocal microscopy; however, the response of the protonated form (with 405 nm excitation) was 9.6-fold less than the response of the anionic form (with 488 nm excitation).

Our group recently published a paper presenting another calcium indicator, named NCaMP7 [9], so we compared the FGCaMP7 indicator to that one. As compared to FGCaMP7, the NCaMP7 indicator [9] belongs to the intensiometric class of GECIs, not ratiometric one. NCaMP7 has also a different design with the insertion of CaM/M13-peptide pair into fluorescent protein; FGCaMP7 has M13-like peptide and CaM attached to the N- and C-terminal ends of the circularly permutated EGFP. NCaMP7 utilizes mNeonGreen fluorescent protein as a fluorescent domain, which provides its enhanced molecular brightness. NCaMP7 has calmodulin similar to that for the GCaMP family, but FGCaMP7 differs in this respect and has calmodulin from fungus, which beneficially ensures its inertness to the intracellular environment in mammalian cells; however, insertion design of NCaMP7 also partially addresses this problem. Finally, as compared to FGCaMP7, NCaMP7 demonstrated a larger response to calcium transients in neurons both in cultured neurons and in vivo.

## Figures and Tables

**Figure 1 ijms-21-03012-f001:**
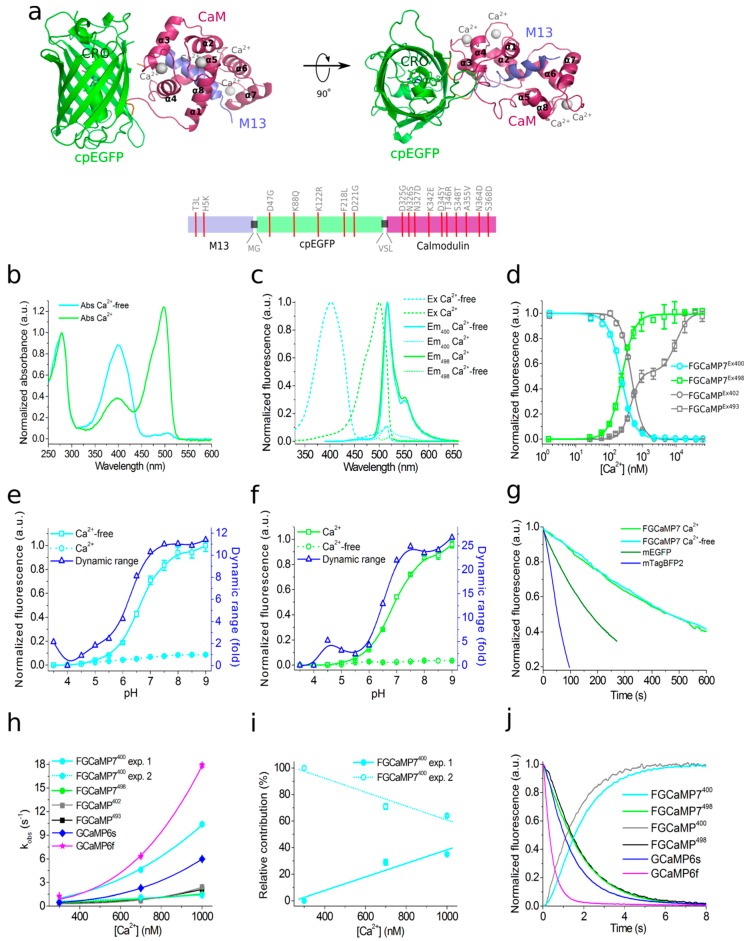
Structure and properties of the FGCaMP7 calcium indicator. (**a**, top) Orthogonal views of the structure of the FGCaMP indicator in Ca^2+^-bound state (PDB 6XU4). β-barrel of cpEGFP is shown as a green cylinder, Ca^2+^ ions are shown as grey spheres, M13-peptide and CaM are shown in purple and light blue, respectively. α-helices of CaM are indicated. (**a**, bottom) Schematic representation of the FGCaMP7 sequence with mutations relative to the original FGCaMP. The numbering follows that for FGCaMP. (**b**) Absorbance spectra in the presence (39 μM) and the absence of Ca^2+^ ions for FGCaMP7. (**c**) Excitation and emission spectra in the presence (39 μM) and the absence of Ca^2+^ ions for FGCaMP7. (**d**) Ca^2+^ titration curves for FGCaMP and FGCaMP7. (**e**) Fluorescence of FGCaMP7 as a function of pH at 365 nm excitation. (**f**) Fluorescence of FGCaMP7 as a function of pH at 490 nm excitation. (**g**) Photobleaching curves for FGCaMP7 in the presence (39 μM) and the absence of Ca^2+^ ions and for mEGFP and mTagBFP2 fluorescent proteins. The power of light before the objective lens was 7.3 mW/cm^2^. (**h**) Observed Ca^2+^-association rate constants determined from association curves for FGCaMP, FGCaMP7, and control GCaMP6s and GCaMP6f. Fast (solid cyan) and slow (dashed cyan) exponents are shown for the FGCaMP7 indicator at 400 nm excitation. Data were fitted to the equation k_obs_ = k_on_ × [Ca^2+^]^n^ + k_off_. (**i**) Relative contribution of monoexponents A_1_/(A_1_ + A_2_) and A_2_/(A_1_ + A_2_) for the FGCaMP7 indicator at 400 nm excitation, where A_1_ and A_2_ are the pre-exponential factors in the association curve equation ΔFlu(t) = A_1_ × exp(-k^on^_obs1_ × t) + A_2_ × exp(-k^on^_obs2_ × t). (**j**) Calcium-dissociation kinetics for FGCaMP, FGCaMP7, GCaMP6s and GCaMP6f. Starting concentration of free Ca^2+^ was 1000 nM. Three replicates were averaged for analysis. Where denoted, whiskers correspond to SD errors.

**Figure 2 ijms-21-03012-f002:**
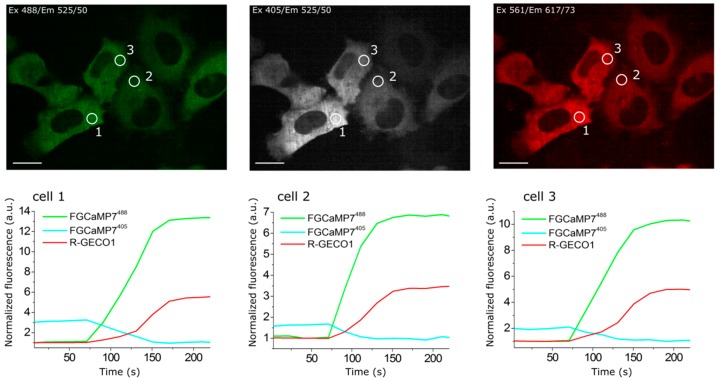
The response of the FGCaMP7 indicator to variations in cytoplasmic calcium concentration in HeLa Kyoto cells. (Upper panels) Confocal images of HeLa Kyoto cells co-expressing ratiometric indicator FGCaMP7 at 488 nm (left panel) and 405 nm (middle panel) excitations and red indicator R-GECO1 at 561 nm excitation (right panel). (Lower panels) The graphs illustrate changes in green fluorescence of FGCaMP7 at 488 nm (green lines) and 405 nm (cyan lines) excitations and red fluorescence of co-expressed R-GECO1 (red lines) GECI in response to 2.5 μM ionomycin. Changes in fluorescence are shown for three cells. The changes shown in graphs correspond to the areas indicated with white circles in the images in upper panels. Scale bar: 20 μm.

**Figure 3 ijms-21-03012-f003:**
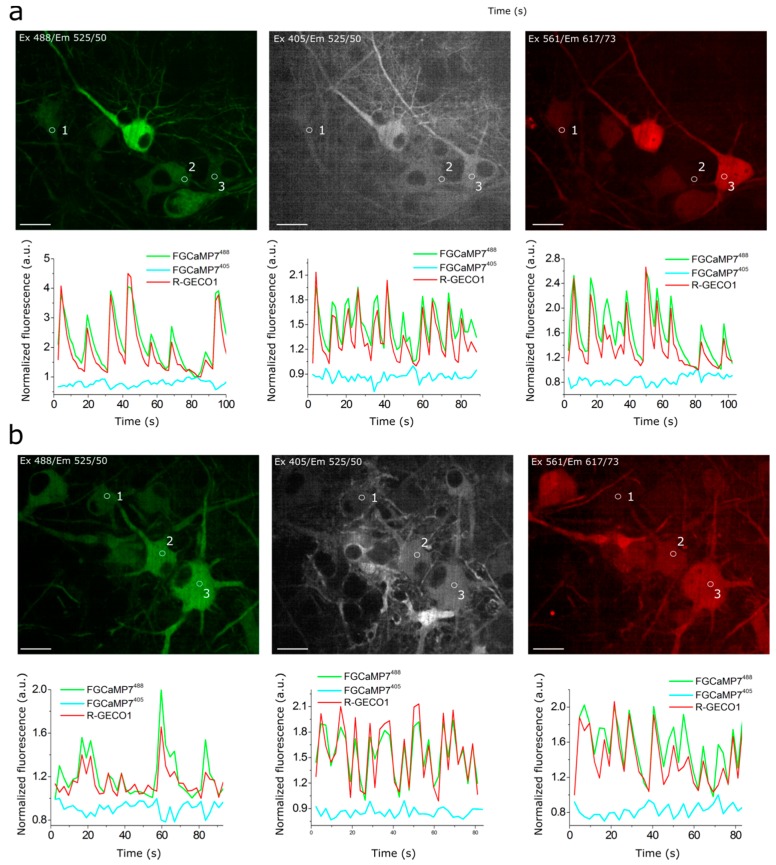
Responses of FGCaMP7 indicator to variations in cytoplasmic calcium concentration in cultured neurons. (**a**,**b**, upper panels) Confocal images of neurons co-expressing ratiometric indicator FGCaMP7 at 488 nm (left panel) and 405 nm (middle panel) excitations and red indicator R-GECO1 at 561 nm excitation (right panel). (**a**,**b**, lower panels) The graphs illustrate changes in green fluorescence of FGCaMP7 at 488 nm (green lines) and 405 nm (cyan lines) excitations and red fluorescence of co-expressed R-GECO1 (red lines) GECI as a result of spontaneous activity in neuronal cultures. Changes in fluorescence are shown for six neurons. The changes shown in graphs correspond to the areas indicated with white circles in the images in upper panels. Scale bar: 20 μm.

**Figure 4 ijms-21-03012-f004:**
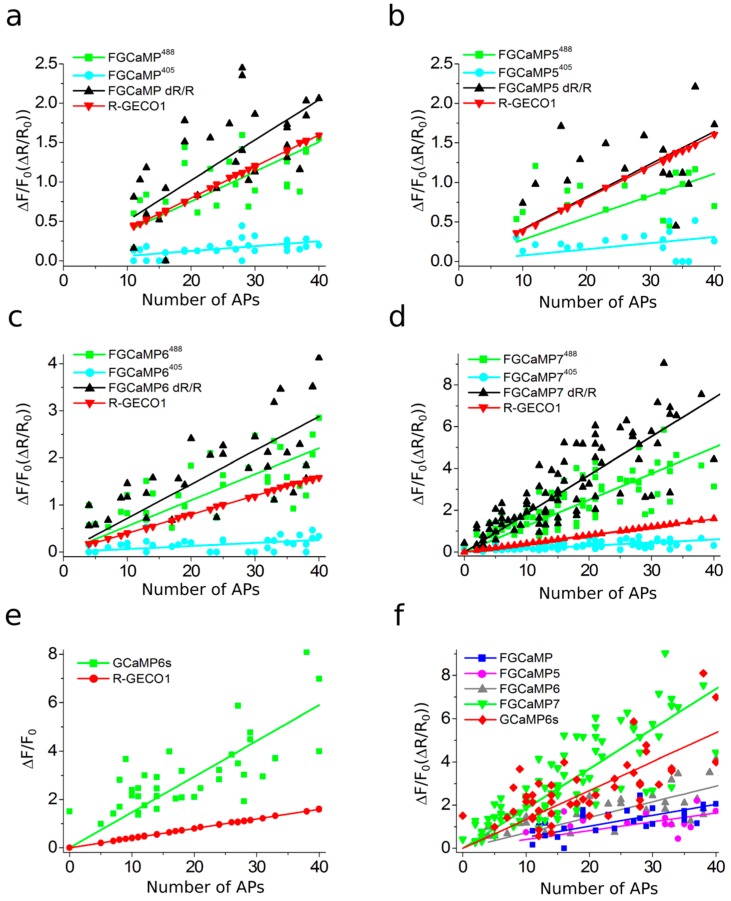
Fluorescence changes in cultured neurons co-expressing FGCaMP series and R-GECO1 in response to electric stimulation. (**a**–**e**) Fluorescence changes (ΔF/F_0_ and ΔR/R_0_) of FGCaMP, FGCaMP5, FGCaMP6, FGCaMP7, and GCaMP6s GECIs in response to external electric stimulation of neuronal cultures as a function of APs. APs were calculated by normalization of ΔF/F_0_ of co-expressed R-GECO1 to the 4% value, which is induced at 1 AP stimuli of R-GECO1, and assuming linear dependence of R-GECO1 response vs. APs [43]. (**f**) Ratiometric changes (ΔR/R_0_) for FGCaMP, FGCaMP5, FGCaMP6, FGCaMP7, and fluorescence change (ΔF/F_0_) for GCaMP6s GECIs.

**Figure 5 ijms-21-03012-f005:**
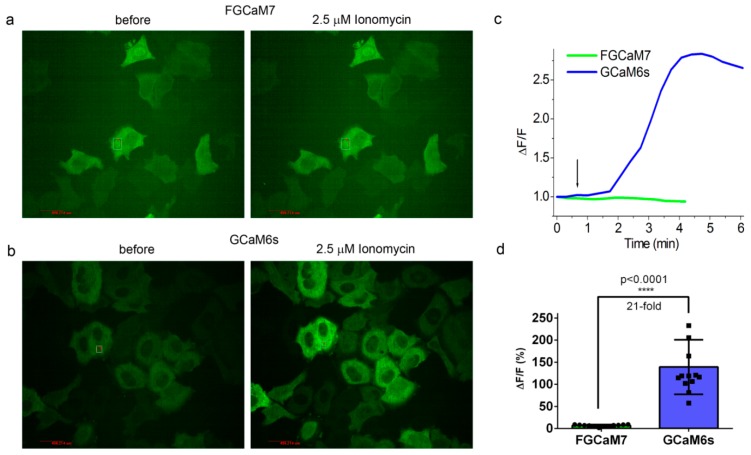
Calcium-dependent response of the truncated versions (with deleted M13-like peptide) of the FGCaMP7 and GCaMP6s indicators in HeLa cells. Confocal images of HeLa cells expressing FGCaM7 (**a**) and GCaM6s (**b**) before and after addition of 2.5 μm ionomycin. (**c**) Graph illustrating calcium-dependent change in ΔF/F_0_ for FGCaM7 (green) and GCaM6s (blue). Addition of 2.5 μm ionomycin is depicted by black arrow. (**d**) Averaged ionomycin-evoked ΔF/F_0_ responses for the FGCaM7 (*n* = 12) and GCaM6s (*n* = 13) trancated indicators.

**Figure 6 ijms-21-03012-f006:**
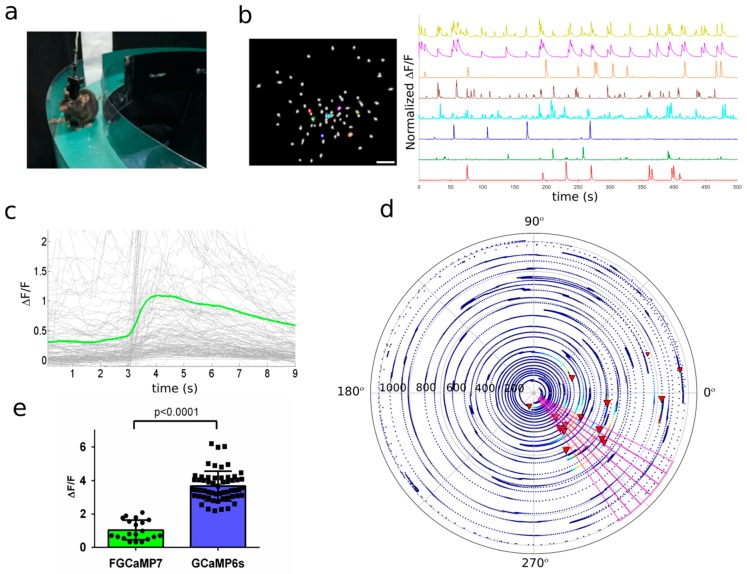
In vivo neuronal Ca^2+^ activity in the hippocampus of freely behaving mice visualized using FGCaMP7 and an nVista HD miniscope. (**a**) Photo of O-shaped track and mouse which explores it with an nVista HD miniscope mounted on its head. (**b**) Spatial filter and sample traces obtained from a 15-min imaging session of freely behaving mouse expressing FGCaMP7 GECI. ΔF/F_0_ values were normalized to the maximal ΔF/F_0_ for each trace. Scale bar: 100 µm. (**c**) Mean spike for FGCaMP7 calcium indicator; spikes above the 4 MAD threshold and not less than 50% of maximal trace value were aligned at the start of the peak (3 s). (**d**) Example of the circular plot for FGCaMP7 mouse trajectory during the exploration of circular track, synchronized with the spikes of a place cell (red triangles). Radial: time, s; angular: intrack position, degrees. (**e**) Averaged ΔF/F_0_ responses for space-evoked activity across place neuronal cells (*n* = 4, FGCaMP7; *n* = 5, GCaMP6s) in the CA1 area of the hippocampus for the FGCaMP7 and GCaMP6s indicators. The FGCaMP7 and GCaMP6s indicators were delivered to the hippocampus with rAAVs carrying AAV-CAG-NES-FGCaMP7/GCaMP6s.

**Figure 7 ijms-21-03012-f007:**
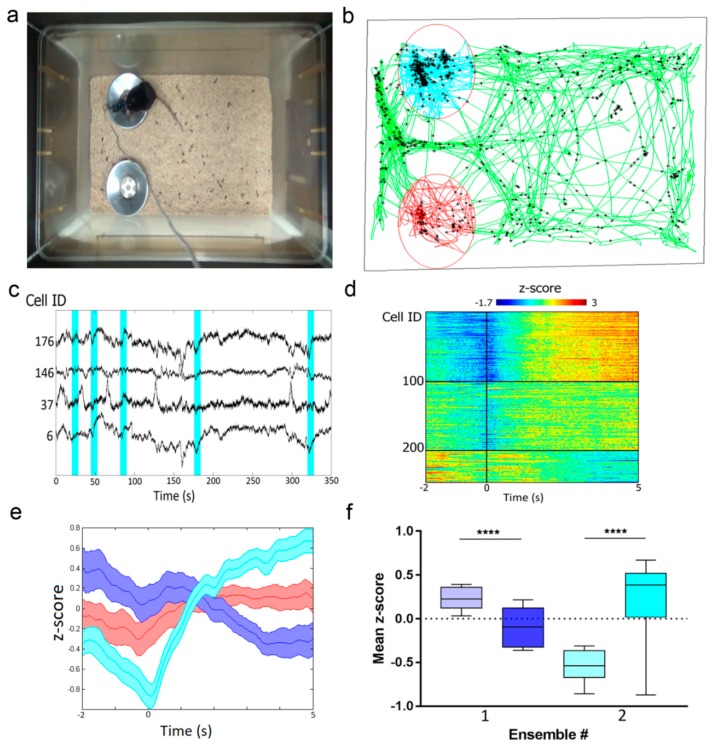
In vivo neuronal Ca2+ activity in mouse hippocampus during food intake visualized using FGCaMP7 and an nVista HD miniscope. (**a**) Photo of the mouse with an nVista HD miniscope mounted on its head during food intake. (**b**) Mouse trajectory for 1-h imaging session (green) with spikes from all detected neurons (black stars), red and cyan curves—trajectory in the area of cups. Context size: 40 cm by 30 cm. (**c**) Example normalized ΔF/F0 Ca2+ signals for four individual neurons recorded during the session. Cyan areas show time windows 2 s before entering the cup area and 5 s after. (**d**) Mean responses of clustered groups of neurons upon cup area-entry transitions (*n* = 253 neurons from two mice; number of entries, 17 and 24). Zero marks cup area-entry time points. Cells were ordered according to k-means clustering. (**e**) Average cup area-entry responses of activated (cyan), inhibited (blue) and indifferent (red) ensembles. Lines indicate the average across neurons ± SEM. (**f**) Average z-score at 2 s time window before entering the cup area and 5 s after for activated (#1) and inhibited (#2) ensembles. *p* values less than 0.0001 are given four asterisks.

**Figure 8 ijms-21-03012-f008:**
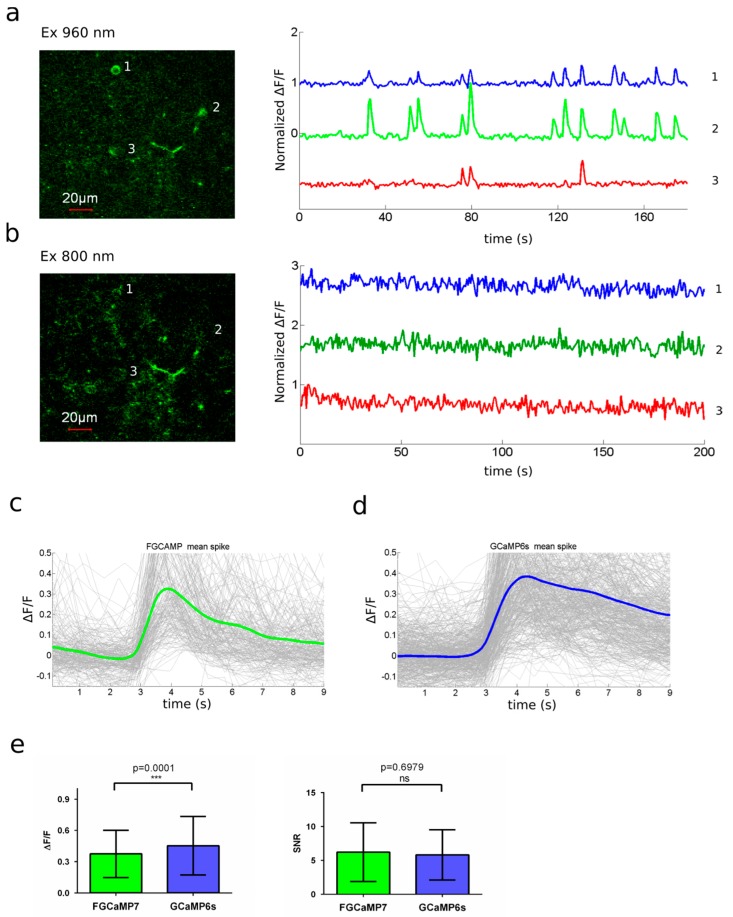
In vivo neuronal Ca^2+^ activity in the visual cortex of awake mice visualized using two-photon microscopy. (**a**,**b**) Two-photon images of V1 layer 2/3 neurons (left) and sample traces for marked neurons (right) acquired during their spontaneous activity at 960 nm (**a**) and 800 nm excitations (**b**) in the mice expressing FGCaMP7 calcium indicator. ΔF/F_0_ value was normalized to maximal ΔF/F_0_ for each trace. (**c**,**d**) Mean spike for the FGCaMP7 and GCaMP6s indicators; spikes with an amplitude above the 3 MAD threshold and not less than 50% of maximal trace value were aligned at the start of the peak (3 s). (**e**) Averaged peak ΔF/F_0_ responses and SNRs for spontaneous neuronal activity (*n* = 20, FGCaMP7; *n* = 14, GCaMP6s) 2/3-layer neurons of visual cortex for the FGCaMP7 and GCaMP6s indicators. *p* value equal to 0.0001 is given three asterisks.

**Figure 9 ijms-21-03012-f009:**
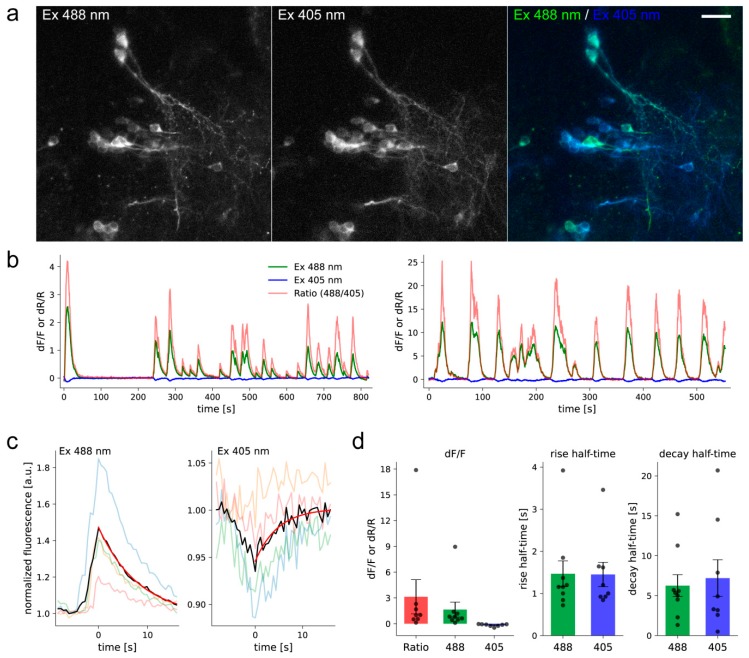
In vivo neuronal Ca^2+^ activity in larval zebrafish brain visualized using laser-scanning confocal microscopy. (**a**) NES-FGCaMP7 transiently expressed in the larval zebrafish brain. Maximum intensity projection image at 488 (left) or 405 (middle) nm excitation, and the merged image (right; 488 in green, 405 in blue). Scale bar: 20 µm. (**b**) Sample of ΔF/F_0_ and ΔR/R_0_ traces of spontaneous activity in 2 cells from 2 zebrafish larvae. Signal obtained at 488 and 405 nm excitation and their ratio (488/405) are shown in green, blue, and red lines, respectively. (**c**) Sample traces of spikes from 1 cell (pale color) and the averaged spike (black). Exponential (red) was fit to the decay of the averaged trace. (**d**) Mean of peak ΔF/F_0_ or ΔR/R_0_ (left), rise half-time (middle), and decay half-time (right) measured on averaged traces for 488 and 405 nm excitation and the ratio (488/405). Each dot represents a value measured in 1 cell. *n* = 9 for 488 nm excitation, 8 for 405 nm excitation and the ratio. Error bars indicate SEM.

**Table 1 ijms-21-03012-t001:** In vitro characterization of FGCaMP7 indicator.

Properties		Proteins			
		FGCaMP		FGCaMP7	
		Apo	Sat	Apo	Sat
Absorbance maximum (nm)		402	493	400	498
Emission maximum (nm)		516		516	
Quantum yield ^a^		0.48 ± 0.02	0.46 ± 0.01	0.40 ± 0.02	0.55 ± 0.04
ε (mM^−1^cm^−1^) ^b^		55 ± 5	106 ± 20	82 ± 11	103 ± 14
Brightness (%) ^c^		100	100	123	116
Fluorescence contrast (fold)	Ex 400/402	6.9 ± 0.5		9.3 ± 0.3	
	Ex 498/493	14.7 ± 0.6		38.4 ± 0.9	
Fluorescence contrast with 1 mM MgCl_2_ (fold)	Ex 400/402	7.1 ± 0.2		10.1 ± 1.0	
	Ex 498/493	15.3 ± 0.5		32.7 ± 1.5	
p*K*_a_ (Ex 400) ^d^		6.56 ± 0.03	7.0 ± 0.6	6.63 ± 0.20	6.00 ± 0.20
p*K*_a_ (Ex 498) ^d^		6.2 ± 0.2	7.33 ± 0.07	5.28 ± 0.10; 7.81 ± 0.10	6.87 ± 0.01
K_d_ (nM) ^e^	Ex 400	276 ± 9 (*n* = 2.8 ± 0.3)		130 ± 10 (*n* = 2.7 ± 0.1)	
	Ex 498	273 ± 7 (*n* = 3.5 ± 0.3) 4700 ± 200 (*n* = 1.9 ± 0.2)		160 ± 10 (*n* = 2.6 ± 0.2)	
K_d_ (nM) with 1 mM MgCl_2_ ^e^	Ex 400	460 ± 60 (*n* = 2.3 ± 0.6)		230 ± 5 (*n* = 2.2 ± 0.1)	
	Ex 498	460 ± 40 (*n* = 2.8 ± 0.4) 4400 ± 800 (*n* = 1.9 ± 0.2)		240 ± 6 (*n* = 2.3 ± 0.2)	
k_obs_ (s^−1^) (300 nM Ca^2+^) ^f^	Ex 400	0.37 ± 0.01		0.60 ± 0.01	
	Ex 498	0.35 ± 0.01		0.42 ± 0.01	
t_1/2_^off^ (s) ^g^	Ex 400	1.2 ± 0.1		1.5 ± 0.1	
	Ex 498	1.4 ± 0.1		1.34 ± 0.02	
Protein state		monomer		monomer	
Maturation half-time (min) ^h^		ND	27 ± 4	ND	ND
Photobleaching half-time (sec) ^i^		54 ± 9	260 ± 40	462 ± 126	464 ± 97

^a^ mEGFP (QY = 0.61 [23]) and mTagBFP2 (QY = 0.64 [27]) were used as reference standards for 493–498- and 400–402-nm absorbing states, respectively. ^b^ Extinction coefficient was determined by alkaline denaturation. ^c^ Brightness normalized to the brightness of the FGCaMP indicator. ^d^ p*K*a values were determined according to the pH dependence of fluorescence. ^e^ Experimental data for FGCaMP7 and 402-form of FGCaMP was fitted to Hill equation, data for 493-form of FGCaMP was fitted to equation y = V_1_*x^n1^/(K_d1_^n1^ + x^n1^) + V_2_*x^n2^/(K_d2_^n2^ + x^n2^). Hill coefficients are shown in brackets. K_d_ for GCaMP6s 144 ± 9 nM (4.0 ± 0.6); in the presence of 1 mM MgCl_2_ K_d_ for GCaMP6s 217 ± 16 nM (4.0 ± 0.6). ^f^ Observed Ca^2+^-associated rate constants were determined from associated curves for FGCaMP and FGCaMP7 in the presence of 1 mM MgCl_2_. k_obs_ for GCaMP6s is 0.49 ± 0.01 s^−1^, k_obs_ for GCaMP6f is 1.28 ± 0.01 s^−1^. ^g^ t_1/2_^off^ values were determined from the dissociation kinetics curves in the presence of 1 mM MgCl_2_. t_1/2_^off^ for GCaMP6s is 1.01 ± 0.06 s; t_1/2_^off^ for GCaMP6f is 0.37 ± 0.04 s. ^h^ mEGFP had a maturation half-time of 13 min. ^i^ mEGFP had a photobleaching half-time of 170 ± 20 s; mTagBFP2 had a photobleaching half-time of 53 ± 9 s.

**Table 2 ijms-21-03012-t002:** Characteristics of FGCaMP, FGCaMP5, FGCaMP6, FGCaMP7, and GCaMP6s GECIs responses to the addition of calcium ions to HeLa cells, to spontaneous activity in dissociated neuronal cultures, and electric stimulation of dissociated neuronal cultures.

		Protein				
		FGCaMP	FGCaMP7	FGCaMP5	FGCaMP6	GCaMP6s
ΔF/F_0_ in HeLa (% vs. R-GECO1)	Ex405	62 ± 22 (*n* = 11)	42 ± 19 *p* = 0.0493 ^a^ (*n* = 10)	44 ± 16 *p* = 0.0243 ^a^ (*n* = 11)	40 ± 22 *p* = 0.0230 ^a^ (*n* = 10)	NA
	Ex488	162 ± 20 *p* = 0.0007 ^b^ (*n* = 11)	237 ± 81 *p* < 0.0001^b^ (*n* = 10)	87 ± 25 *p* = 0.0305 ^b^ (*n* = 11)	160 ± 55 *p* = 0.0297 ^b^ (*n* = 10)	113 ± 35 (*n* = 15)
ΔF/F_0_ in neurons (% vs. R-GECO1)	Ex405	50 ± 20 (*n* = 12)	34 ± 13 *p* = 0.0267 ^a^ (*n* = 11)	19 ± 10 *p* = 0.0014 ^a^ (*n* = 11)	46 ± 37 *p* = 0.2412 ^a^ (*n* = 13)	NA
	Ex488	150 ± 53 *p* = 0.3299 ^b^ (*n* = 12)	142 ± 35 *p* = 0.4966 ^b^ (*n* = 10)	58 ± 25 *p* < 0.0001 ^b^ (*n* = 16)	143 ± 40 *p* = 0.3932 ^b^ (*n* = 14)	133 ± 58 (*n* = 17)
ΔR/R_0_ ^c^ in neurons (% vs. R-GECO1)		245 ± 110 *p* = 0.0030 ^b^ (*n* = 11)	219 ± 78 *p* = 0.0014 ^b^ (*n* = 10)	103 ± 42 *p* = 0.1436 ^b^ (*n* = 11)	231 ± 70 *p* = 0.0003 ^b^ (*n* = 13)	NA
rise half-time (s) ^d^	Ex405	1.93 ± 0.62	1.36 ± 0.47	2.45 ± 1.09	2.01 ± 0.93	NA
	Ex488	2.08 ± 0.57	1.36 ± 0.47	2.01 ± 1.23	1.65 ± 0.71	1.14 ± 0.61
decay half-time (s) ^e^	Ex405	3.56 ± 1.37	3.40 ± 0.72	5.78 ± 2.56	4.41 ± 2.18	NA
	Ex488	4.01 ± 1.65	3.84 ± 0.88	6.51 ± 2.50	4.89 ± 1.92	3.77 ± 1.27
ΔF/F_0_ per 1 AP in stimulated neurons (%)	Ex405	0.6 ± 0.4 ^f^ (*n* = 29)	2.1 ± 1.5 ^g^ *p* < 0.0001 ^a^ (*n* = 87)	1.1 ± 0.8 ^f^ (*p* = 0.0148) ^a^ (*n* = 17)	0.9 ± 0.9 ^h^ (*p* = 0.2678) ^a^ (*n* = 34)	NA
	Ex488	4.1 ± 1.7 ^f^ *p* < 0.0001 ^b^ (*n* = 29)	15.0 ± 7.5 ^g^ (*p* = 0.8805) ^b^ (*n* = 90)	3.5 ± 1.6 ^f^ *p* < 0.0001 ^b^ (*n* = 17)	7.3 ± 4.2 ^h^ *p* < 0.0001 ^b^ (*n* = 34)	15.0 ± 7.0 ^g^ (*n* = 50)
ΔR/R_0_ ^c^ per 1 AP in stimulated neurons (%)		5.4 ± 2.3 ^f^ *p* < 0.0001 ^b^ (*n* = 29)	20.3 ± 9.2 ^g^ (*p* = 0.0002) ^b^ (*n* = 86)	5.1 ± 2.4 ^f^ *p* < 0.0001 ^b^ (*n* = 17)	9.2 ± 4.7 ^h^ *p* < 0.0001 ^b^ (*n* = 34)	NA

^a^*p* value shows the statistical difference vs. the FGCaMP indicator. ^b^
*p*-value shows the statistical difference vs. GCaMP6s. ^c^ ΔR/R_0_ was measured as (I_488_(Ca^2+^)/I_405_(Ca^2+^) − I_488_(noCa^2+^)/I_405_(noCa^2+^))/I_488_(noCa^2+^)/I_405_(noCa^2+^), where I_488_(Ca^2+^) and I_405_(Ca^2+^) are intensities of the indicator under Ca^2+^-elevated conditions during spontaneous activity or in the response to electric field stimulation at 488 or 405 nm excitation, while I_488_(noCa^2+^) and I_405_(noCa^2+^) are intensities during Ca^2+^ depletion under the same conditions. ^d^ Rise half-time was measured as the time between the onset of neuronal activity and the half-peak of the response. ^e^ Decay half-time was measured as the time from the peak to the half-peak at the end of the response. ^f^ ΔF/F_0_ or ΔR/R_0_ per 1 AP was calculated as a slope of linear function ΔF/F_0_ from 10–40 APs. ^g^ ΔF/F_0_ or ΔR/R_0_ per 1 AP was calculated as a slope of linear function ΔF/F_0_ from 1–40 APs. ^h^ ΔF/F_0_ or ΔR/R_0_ per 1 AP was calculated as a slope of linear function ΔF/F_0_ from 4–40 APs. NA, not applicable.

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
