# Peer review of "FGCaMP7, an Improved Version of Fungi-Based Ratiometric Calcium Indicator for In Vivo Visualization of Neuronal Activity"

_ijms, 2020, doi:10.3390/ijms21083012_

Round 1
Reviewer 1 Report
This paper presents improvement on a calcium indicator, first published in 2017 (ref 19). The data is presented in a logical progression, but it is hard to read. The paper is interesting, but the presentation could be improved.
Issues:
Figure S2 b, top, how were spectra normalized?
The section on the initial directed mutagenesis (lines 192-260) could be more concise. It’s interesting data, but ultimately did not lead to a better indicator.
The Kd1 values given in the text (lines 283-284) do not match the numbers in Fig. S3 for FGCaMP2 and 3. This needs to be corrected (although these mutants were not selected for further work).
For directed evolution of FGCaMP4, the authors state:
During the first step, we performed imaging of the indicator’s library
295 to the E. coli periplasm on Petri dishes and selected clones with the highest fluorescence ratio before
296 and after treatment with a buffer that contained ethylenediaminetetraacetic acid (EDTA) using
297 405/40BP and 480/40BP nm excitation filters and 510LP and 535/40BP nm emission filters,
298 respectively.
Why this choice of emission filters? 535/40BP range is within 510LP range.
Line 421: upper value for FGCaMP5 is missing
The paper needs English editing throughout, there are many grammar errors. Here are just a few examples (there are many!):
- many prepositions are incorrect
- In some extent should be to some extent
- Access should be assess in lines 361 and 633
- In the same conditions should be under the same conditions
This group recently published a paper presenting another calcium indicator (ref. 9), how does this new indicator compare to that one?
Author Response
Response to Reviewer 1 Comments
We thank reviewer 1 for his valuable comments and useful suggestions, which we have addressed entirely in the revised manuscript.
Reviewer #1:
This paper presents improvement on a calcium indicator, first published in 2017 (ref 19). The data is presented in a logical progression, but it is hard to read. The paper is interesting, but the presentation could be improved.
Issues:
Point 1: Figure S2 b, top, how were spectra normalized?
Response 1: In the revised manuscript, Supplementary Information, legend to the Figure S2b, top, we added “Absorbance at 280 nm is normalized to 1.”.
Point 2: The section on the initial directed mutagenesis (lines 192-260) could be more concise. It’s interesting data, but ultimately did not lead to a better indicator.
Response 2: In the revised manuscript we would like to keep this section since we describe the structural effects of those mutations that led to changes in the properties of the FGCaMP protein. The main goal in this section was not to improve the performance of the FGCaMP indicator, but to confirm and study the FGCaMP X-ray structure using point mutations.
Point 3: The Kd1 values given in the text (lines 283-284) do not match the numbers in Fig. S3 for FGCaMP2 and 3. This needs to be corrected (although these mutants were not selected for further work).
Response 3: In the revised manuscript, Supplementary Information, on Figure S3 we provided correct Kd1 values for the FGCaMP2/T28D (Kd1 433 nM) and FGCaMP3 (Kd1 208 nM) indicators.
Point 4: For directed evolution of FGCaMP4, the authors state:
During the first step, we performed imaging of the indicator’s library
295 to the E. coli periplasm on Petri dishes and selected clones with the highest fluorescence ratio before
296 and after treatment with a buffer that contained ethylenediaminetetraacetic acid (EDTA) using
297 405/40BP and 480/40BP nm excitation filters and 510LP and 535/40BP nm emission filters,
298 respectively.
Why this choice of emission filters? 535/40BP range is within 510LP range.
Response 4: We used these emission filters for convenient registration of green fluorescence using two filter sets. So both 535/40BP and 510LP filters worked well, with no difference. There was no choice to prefer a particular emission filter.
Point 5: Line 421: upper value for FGCaMP5 is missing
Response 5: In the revised manuscript, Main text, page 10, last paragraph, we replaced “…varied from 10 to for FGCaMP5…” with “…varied from 10 to 29% for FGCaMP5…”.
Point 6: The paper needs English editing throughout, there are many grammar errors. Here are just a few examples (there are many!):
many prepositions are incorrect
In some extent should be to some extent
Access should be assess in lines 361 and 633
In the same conditions should be under the same conditions
Response 6: In the revised manuscript, main text, page 3, 1st paragraph, we replaced “…demonstrated that topology changed in this way may to some extent prevent calcium indicator interactions with intracellular environment…” with “…demonstrated that, when the topology is changed in this way, calcium indicator interactions with intracellular environment may, to some extent, be prevented…”.
In the revised manuscript, main text, page 9, 2nd paragraph, we replaced “We next accessed the affinity…” with “We next assessed the affinity…”.
In the revised manuscript, main text, page 15, 3rd paragraph, we replaced “…in vivo, we accessed its response…” with ““…in vivo, we assessed its response…”
In the revised manuscript, main text, page 9, 2nd paragraph, we replaced “…for FGCaMP in the same conditions and were optimal…” with ““…for FGCaMP under the same conditions and were optimal…”
In the revised manuscript, main text, page 10, 1st paragraph, we replaced “…FGCaMP6 were negligible in the same conditions …” with ““…FGCaMP6 were negligible under the same conditions …”
To address this point entirely we corrected English using the MDPI English editing service.
Point 7: This group recently published a paper presenting another calcium indicator (ref. 9), how does this new indicator compare to that one?
Response 7: In the revised manuscript, Main text, Conclusion section, we added: “Our group recently published a paper presenting another calcium indicator, named NCaMP7 [9], so we compared the FGCaMP7 indicator to that one. As compared to FGCaMP7, the NCaMP7 indicator [9] belongs to intensiometric class of GECIs, not ratiometric one. NCaMP7 has also different design with insertion of CaM/M13-peptide pair into fluorescent protein; FGCaMP7 has M13-like peptide and CaM attached to the N- and C-terminal ends of the circularly permutated EGFP. NCaMP7 utilizes mNeonGreen fluorescent protein as fluorescent domain, which provides its enhanced molecular brightness. NCaMP7 has calmodulin similar to that for GCaMP family, but FGCaMP7 differs in this respect and has calmodulin from fungus, which beneficially ensures its inertness to the intracellular environment in mammalian cells; however, insertion design of NCaMP7 also partially addresses this problem. Finally, as compared to FGCaMP7, NCaMP7 demonstrated larger response to calcium transients in neurons both in cultured neurons and in vivo.”
Reviewer 2 Report
In this nice article, Barykina and others describe the development and characterization of a novel genetically encoded calcium sensor, named FGCaMP7, based on the utilization of orthogonal CaM and M13 components (originated from fungi) so that they reduce to a minimum the interference with endogenous cellular processes (e.g. intracellular calcium homeostasis, CaV-mediated cellular excitability, etc.).
Overall the results are novel and interesting, especially given the presentation of in vivo data both in mice and fishes that nicely demonstrate the functionality of the probe in a complete living system. While the response amplitudes and contrast of FGCaMP7 for in vivo imaging are still lower than those of the widely used GCaMP6, they may represent a good starting point for further optimization of the probe.
Comments for improvement:
Given that the most important advantage of FGCaMP7 compared to other existing GECIs is the reduced interference with intracellular systems, I feel that the authors should make an extra effort to demonstrate that. The only figure dealing with this aspect of the results is Figure 5, which nicely shows how GCaMP-type sensors can indeed rely on interactions with endogenous proteins for their fluorescent responses. However, it would be beneficial to the paper if the authors could show what are the different effects on cellular toxicity of the two probes: FGCaMP7 and GCaMP6. For instance it would be interesting to know whether upon prolonged expression times in vivo GCaMP6 leads to a higher amount of intracellular aggregates than FGCaMP7.
Separate from the issue of interacting with endogenous proteins, GECIs are also known to cause the issue of calcium buffering: https://www.cell.com/trends/neurosciences/pdf/S0166-2236(18)30247-9.pdf
It would be great if the authors could comment on whether their new sensors could also bypass some of these side-effects or not.
A main application of GECIs is with in vivo photometry, and an advantage of GCaMP6 in that regard is the possibility to use 405nm as an 'isosbestic' wavelength, where the fluorescence emission is not affected by calcium. This is typically used to normalize signals and remove motion artefacts. The authors should comment in the Conclusions of the article on the relative differences of their new probe in these regards.
Author Response
Response to Reviewer 2 Comments
We thank reviewer 2 for his valuable comments and useful suggestions, which we have addressed entirely in the revised manuscript.
Reviewer #2:
In this nice article, Barykina and others describe the development and characterization of a novel genetically encoded calcium sensor, named FGCaMP7, based on the utilization of orthogonal CaM and M13 components (originated from fungi) so that they reduce to a minimum the interference with endogenous cellular processes (e.g. intracellular calcium homeostasis, CaV-mediated cellular excitability, etc.).
Overall the results are novel and interesting, especially given the presentation of in vivo data both in mice and fishes that nicely demonstrate the functionality of the probe in a complete living system. While the response amplitudes and contrast of FGCaMP7 for in vivo imaging are still lower than those of the widely used GCaMP6, they may represent a good starting point for further optimization of the probe.
Comments for improvement:
Point 1: Given that the most important advantage of FGCaMP7 compared to other existing GECIs is the reduced interference with intracellular systems, I feel that the authors should make an extra effort to demonstrate that. The only figure dealing with this aspect of the results is Figure 5, which nicely shows how GCaMP-type sensors can indeed rely on interactions with endogenous proteins for their fluorescent responses. However, it would be beneficial to the paper if the authors could show what are the different effects on cellular toxicity of the two probes: FGCaMP7 and GCaMP6. For instance it would be interesting to know whether upon prolonged expression times in vivo GCaMP6 leads to a higher amount of intracellular aggregates than FGCaMP7.
Response 1: In the revised manuscript, Main text, page 12, we added: “Both FGCaMP7 and GCaMP6s indicators demonstrated even distribution in the cytosol of neurons, however R-GECO1 indicator revealed puncta-like aggregates during prolonged expression in neuronal cultures (Figure S11). This difference may be attributed to different fluorescent domains utilized in FGCaMP7 and GCaMP6s (EGFP fluorescent domain) as compared to R-GECO1 (mApple fluorescent domain). Puncta formation was attributed to lysosomes [40], therefore autophagy should adjust the number of these puncta [41].”
In the revised manuscript, Main text, page 15, we added: “We could not reveal a noticeable difference between the FGCaMP7 and GCaMP6s indicators according to the appearance of cells having an unusual shape and intracellular aggregates even upon prolonged in vivo expression of these indicators in hippocampus of mice (Figure S12). A separate study with better microscopic resolution and more statistics is needed to identify possible difference in cellular toxicity between FGCaMP7 and GCaMP6s.”
Point 2: Separate from the issue of interacting with endogenous proteins, GECIs are also known to cause the issue of calcium buffering: https://www.cell.com/trends/neurosciences/pdf/S0166-2236(18)30247-9.pdf
It would be great if the authors could comment on whether their new sensors could also bypass some of these side-effects or not.
Response 2: In the revised manuscript, Main text, Conclusions section, we added: “Similarly to GCaMPs, FGCaMP7 has four calcium binding sites per one molecule. Hence, its overexpression in cells may affect calcium ions concentration in the same way as GCaMPs [62]. Hence, FGCaMP7 does not have advantage in the respect of calcium buffering.”.
Point 3: A main application of GECIs is with in vivo photometry, and an advantage of GCaMP6 in that regard is the possibility to use 405nm as an 'isosbestic' wavelength, where the fluorescence emission is not affected by calcium. This is typically used to normalize signals and remove motion artefacts. The authors should comment in the Conclusions of the article on the relative differences of their new probe in these regards.
Response 3: In the revised manuscript, Main text, Conclusion section, we added: “The main application of GECIs is in vivo photometry, and the advantage of the non-ratiometric GCaMP6 indicators in this regard is the ability to use 405 nm as the “isobestic” wavelength, where calcium does not affect fluorescence [62, 63]. This is typically used to normalize signals and remove motion artefacts. In this type of applications, the control 405 nm light should be avoided in the case of ratiometric FGCaMP7 and other control wavelengths such as 561 or 640 nm might be used.”.